# Cancer-associated mutations in DICER1 RNase IIIa and IIIb domains exert similar effects on miRNA biogenesis

Jeffrey Vedanayagam [1], Walid K. Chatila[2,3,4], Bülent Arman Aksoy [2,3,10], Sonali Majumdar[1], Anders Jacobsen Skanderup [2,11], Emek Demir[2,5], Nikolaus Schultz[4,6,7], Chris Sander [2,8,9] & Eric C. Lai [1,3]

Somatic mutations in the RNase IIIb domain of DICER1 arise in cancer and disrupt the cleavage of 5' pre-miRNA arms. Here, we characterize an unstudied, recurrent, mutation (S1344L) in the DICER1 RNase IIIa domain in tumors from The Cancer Genome Atlas (TCGA) project and MSK-IMPACT profiling. RNase IIIa/b hotspots are absent from most cancers, but are notably enriched in uterine cancers. Systematic analysis of TCGA small RNA datasets show that DICER1 RNase IIIa-S1344L tumors deplete 5p-miRNAs, analogous to RNase IIIb hotspot samples. Structural and evolutionary coupling analyses reveal constrained proximity of RNase IIIa-S1344 to the RNase IIIb catalytic site, rationalizing why mutation of this site phenocopies known hotspot alterations. Finally, examination of DICER1 hotspot endometrial tumors reveals derepression of specific miRNA target signatures. In summary, comprehensive analyses of DICER1 somatic mutations and small RNA data reveal a mechanistic aspect of pre-miRNA processing that manifests in specific cancer settings.

[1] Department of Developmental Biology, Memorial Sloan-Kettering Cancer Center, New York, NY 10065, USA. [2] Department of Computational Biology, Memorial Sloan-Kettering Cancer Center, New York, NY 10065, USA. [3] Tri-Institutional Program in Computational Biology and Medicine, Weill Cornell Medical College, New York, NY 10065, USA. [4] Marie-Josee and Henry R. Kravis Center for Molecular Oncology, Memorial Sloan Kettering Cancer Center, New York, NY 10065, USA. [5] Oregon Health and Science University, Computational Biology Program, Portland, OR 97239, USA. [6] Human Oncology and Pathogenesis Program, Memorial Sloan Kettering Cancer Center, New York, NY 10065, USA. [7] Departments of Epidemiology and Biostatistics, Memorial Sloan Kettering Cancer Center, New York, NY 10065, USA. [8] cBio Center, Dana-Farber Cancer Institute, Boston, MA 02115, USA. [9] Department of Cell Biology, Harvard Medical School, Boston, MA 02115, USA. [10]Present address: Immunology and Microbiology Department, Medical University of South Carolina, Charleston, SC 29412, USA. [11]Present address: Computational and Systems Biology, Agency for Science Technology and Research, Genome Institute of Singapore, 60 Biopolis Street, Singapore 138672, Singapore. Correspondence and requests for materials should be addressed to C.S. (email: sander.research@gmail.com) or to E.C.L. (email: laie@mskcc.org)

MicroRNAs (miRNAs) are ~22 nucleotide (nt) RNAs that mediate post transcriptional repression in diverse species[1]. In animals, most miRNAs traverse a canonical biogenesis pathway involving compartmentalized processing by two RNase III enzymes[2]. In the nucleus, primary miRNA transcripts bearing inverted repeats are cleaved by Drosha to release pre-miRNA hairpins. In the cytoplasm, these are cleaved by Dicer to yield miRNA/miRNA* duplexes, which load into Argonaute effector proteins[3]. Following removal of miRNA* species, the single-stranded miRNA-Argonaute complex, in association with GW182/TNRC6 cofactors, seeks regulatory targets. In addition, diverse non-canonical biogenesis pathways that bypass Drosha and/or Dicer can also generate functional Argonaute-loaded miRNAs[2].

The most critical determinant for target recognition is Watson–Crick base-pairing to the miRNA seed region, nts 2–8 from the 5′ end[1,4]. Owing to—or perhaps facilitated by—the modest amount of sequence complementarity needed for miRNA regulation, most animal miRNAs appear to have been incorporated into large target networks. There is both computational[5] and experimental[6,7] evidence that individual miRNAs can regulate hundreds of genes. Although it remains challenging to reconcile the evidence for such broad miRNA regulatory networks with the often nominal defects observed in individual miRNA knockouts[8], documented miRNA mutants exhibit developmental, physiological, metabolic, and/or behavioral defects[9]. Moreover, null mutants in core miRNA biogenesis factors are lethal in all animals[10–13], and yield severe tissue-specific defects when inactivated conditionally[10,14–16]. Strikingly, recent studies reveal that mutation of human DICER1 is cell lethal in human embryonic stem cells[17]. This serves as a testament to the detrimental impact of ablating miRNA-mediated regulation.

Surprisingly, then, mutations in human DICER1 are recurrent in diverse cancers[18–24]. Thus, although miRNAs are required for normal cell fitness, selective inactivation of DICER1 can benefit cancer cells. DICER1 hotspot mutations occur preferentially within the RNase IIIb domain, and usually affect the metal ion-binding residues[25]. Notably, mechanistic studies showed that activity of the Dicer RNase III domains can be uncoupled[26,27]. In particular, RNase IIIa cuts the 3p hairpin arm, while RNase IIIb cuts the 5p hairpin arm. Accordingly, cancer hotspot mutant variants of DICER1 exhibit selective defects in processing miRNA-5p strands, leading to overall decreases in 5p:3p strand ratios[20,22–24,26].

DICER1 hotspot mutants were mostly characterized using cell models and in vitro assays, but consequences of biased miRNA processing on target regulation have not been studied extensively within human tumors. For example, the relative contribution of 5p-strand depletion or increased 3p-strand accessibility in cancer remains to clarified, as well as if particular miRNAs or families drive the phenotype. In addition, although large numbers of somatic DICER1 mutations appear in cancer genome sequencing, it is relevant to know if additional driver alleles beyond RNase IIIb hotspots can be distinguished.

Here, we perform integrative studies on genome sequence, small RNA and RNA-seq data from the Cancer Genome Atlas (TCGA) PanCancer (PanCan) project[28,29] [https://portal.gdc.cancer.gov/] and Memorial Sloan-Kettering Integrated Mutation Profiling of Actionable Cancer Targets (MSK-IMPACT) clinical profiling[30] [https://www.mskcc.org/msk-impact]. These analyses yield insights into tumor-specificity of DICER1 mutations and their consequences on miRNA and mRNA profiles. Surprisingly, we reveal that recurrent DICER1 RNase IIIa-S1344L mutants impede miRNA-5p biogenesis and activity, similar to known RNase IIIb mutants. Evolutionary and structural evidence shows that RNase IIIa-S1344 is functionally coupled to RNase IIIb

catalytic residues. Finally, we exploit prevalent DICER1 RNase III hotspot mutations in endometrial cancer to identify targets of specific miRNAs that are derepressed in this setting.

## Results

**Recurrent DICER1 RNase IIIa and IIIb mutations in tumors.** Ever since the finding that certain tumors accumulate DICER1 lesions, especially point mutations within its RNase IIIb domain, many studies analyzed DICER1 alleles in diverse cancers and pathologies[25]. Here, we summarize the landscape of somatic DICER1 mutations from 9919 TCGA PanCan datasets[28,29], and 31,029 IMPACT datasets acquired from targeted sequencing of MSK patients[30]. While TCGA data comprises primary tumors from untreated patients, IMPACT systematically profiles all MSK patients, and includes advanced and metastatic cancers. Thus, we have the opportunity to compare features of DICER1 mutations between these large and functionally independent cohorts.

While overall infrequent, the known RNase IIIb mutations comprise clear hotspots in these cross-cancer profilings (Fig. 1a). Mutations in the metal ion-binding residues at the catalytic center (E1705, D1709, D1810, and E1813) constitute the most frequent hits in RNase IIIb. Cancer-associated mutation of G1809 partially impairs RNase IIIb activity[31]; this allele was present only once in the TCGA dataset but was clearly recurrent in the IMPACT data with eight cases. There were 79 hits in these five residues across the two datasets (Fig. 1a). More recently, mutation of RNase IIIb-D1713 was reported in anaplastic kidney sarcoma and shown to impair RNase IIIb activity[23]. However, only a single case occurred in TCGA data and none appeared in IMPACT (Supplementary Data 1 and 2); thus, we did not consider this a hotspot.

In stark contrast to these RNase IIIb mutations, there is a paucity of RNase IIIa mutations affecting catalysis. Despite 845 somatic hits in the aggregate DICER1 data, three of the RNase IIIa metal-binding residues (E1316, D1561, and E1564) were never mutated, and only single instances affected the fourth residue (D1320N) in TCGA and IMPACT data (Supplementary Data 1 and 2). Thus, there does not appear to be any selective advantage to inactivate DICER1 RNase IIIa in cancer. Nevertheless, we were intrigued by a recurrent mutation in RNase IIIa (S1344L, and one instance of S1344T), represented in both TCGA and IMPACT cohorts (Fig. 1a). Two prior studies each recorded single instances of S1344L in Wilms' tumor[20,32], although no specific tests of this allele were reported. Interestingly, we observed that among TCGA cases, alterations of S1344 were similar in frequency to individual RNase IIIb catalytic site mutants. We also note some other uncharacterized somatic mutations shared by both datasets, including ones affecting the PAZ domain (R944Q) and RNase IIIb domain (D1699N/D/Xsplice). Notably, while only RNase IIIb mutations were previously classified as hotspots[33], accumulating tumor sequencing data now enables classification of RNase IIIa-S1344L/T as a statistically significant recurrent cancer mutation[34]. The current iteration of Cancer Hotspots analysis places a q-value for S1344L/T (0.0281) on par with RNase IIIb-D1709N (0.0130), although q-values for other RNase IIIb hotspots are lower (e.g., D1810 mutations, 8.22E-5; E1813 mutations, 1.04E-10; Supplementary Fig. 1). No other DICER1 residues are currently imputed as statistically significant recurrent targets (https://www.cancerhotspots.org).

**Biallelic DICER1 cases involve RNase IIIa and IIIb hotspots.** Evidence was reported from mouse models that Dicer is haploinsufficient in certain cancer contexts[35,36], and heterozygous germline mutations in DICER1 were first detected in pleuropulmonary blastoma[18]. However, other studies indicate that

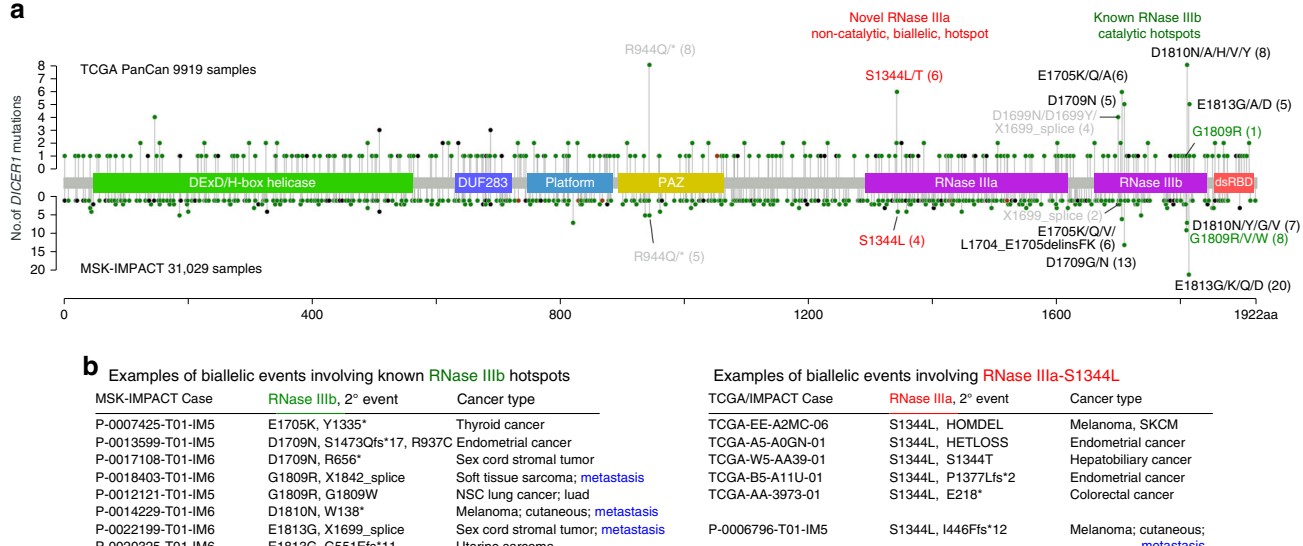

**Fig. 1** Survey of *DICER1* cancer mutations reveals recurrent RNase IIIb and IIIa hotspots. **a** Lollipop mutation diagrams of TCGA-PanCan (pointing up) and MSK-IMPACT (pointing down) datasets. The sites labeled in black designate mutations in the four catalytic residues in the RNase IIIb domain (D1709, E1705, E1813, D1810, in black) and an adjacent residue (G1809, in green) that are the major known biallelic *DICER1* events in cancer. These aggregate data also reveal an uncharacterized biallelic alteration involving S1344L (in red) in the RNase IIIa domain. The background of other somatic mutations of unknown functional significance is shown for perspective; certain other apparently recurrent mutations are recovered in both TCGA and IMPACT datasets (e.g., PAZ-R944Q, in gray), but were not associated with biallelic hits in non-hypermutated samples. **b** Left column, examples of biallelic hits in each of the five known RNase IIIb hotspot residues in IMPACT data, with their corresponding secondary inactivating mutations and cancer types. Right column, a selection of multiple cancer types that exhibit RNase IIIa-S1344L and carry diverse secondary inactivating mutations across TCGA and IMPACT data. Patient sequencing from metastatic tumors are noted (in blue)

*DICER1* hotspot mutations are biallelic in cancer, and act in trans to nonsense or inactivating alleles of DICER1[20,21,37]. Surprisingly, certain murine cancer models are supported[38] or indeed driven[39] by full conditional inactivation of *Dicer*, suggesting that it can act as a conventional tumor suppressor in certain contexts. This may be the case in human pineoblastoma, for which germline *DICER1* mutation combined with loss-of-heterozygosity was detected[40].

To clarify these genetic observations with respect to human tumor data, we sought the existence of secondary disabling events in *DICER1* hotspot cases. There are caveats to this analysis, such as the inference that secondary aberrations occur in trans as opposed to in cis, and whether they occur within the same nuclei as opposed to different tumor cells. Moreover, it is more challenging to call indels and deletions compared to point mutations, and IMPACT data are generally underpowered relative to TCGA data for calling copy number variations (CNVs). Nevertheless, we used the latest TCGA MC3 MAF files using 2+ callers[29] to screen all somatic *DICER1* missense mutant tumors for overt secondary disabling events (nonsense mutations, out-of-frame indels, splice-altering alleles or deep deletions). This approach would miss secondary events called by only one algorithm, but those events called are considered robust.

For initial assessments, we excluded hypermutated tumors (mostly POLE and some MSI), which are expected to exhibit elevated passenger mutations. For example, the recurrent allele R944Q (rarely R944*) could be found with heterozygous copy loss, or with other somatic *DICER1* variants of unknown significance, but all of these were strictly found in hypermutated tumors. In general, apparent passenger mutations in *DICER1* were enriched in hypermutated cases (mostly colorectal and esophageal cancers, and POLE subclass of endometrial cancers) in both TCGA and IMPACT cohorts (Fig. 2 and Supplementary Data 1 and 2).

Overall, this approach yielded a clarified picture. While the vast majority of somatic mutations fell away, many RNase IIIb mutant

tumors contained biallelic events (Supplementary Data 1 and 2). We highlight examples of biallelic events involving each of the five RNase IIIb DICER1 hotspot residues among IMPACT tumors (Fig. 1b). These data support the notion that *DICER1* is not generally haploinsufficient in human cancer, but instead requires a cellular environment where only an altered Dicer activity is present. We subsequently recognized additional biallelic cases involving RNase IIIb hotspot mutations in hypermutated samples (Supplementary Data 1 and 2), suggesting that functional alteration of DICER1 is also relevant in these patients. By contrast, hardly any other alleles exhibited evidence for biallelic events that were recurrent in both datasets. One possible example was D1699N (Fig. 1a), which was picked up once in a HETLOSS tumor and another time as a splice-inactivating event with an RNase IIIb hotspot (E1813G); other instances of this allele occurred in hypermutated tumors.

Of perhaps greater interest, the only uncharacterized mutation with clearly recurrent biallelic events in both non-hypermutated TCGA and IMPACT datasets was RNase IIIa-S1344L (ten events in nine patients, Fig. 1a and Supplementary Data 1 and 2). Figure 1b highlights S1344L events in trans to deletions, nonsense mutations, and an exceptional case bearing a second S1344 allele (S1344T). While prior genome sequencing has not implicated RNase IIIa mutations as pathogenic, and RNase IIIa catalytic site mutations are indeed nearly absent in cancer, these observations further suggest S1344L is functionally relevant during tumorigenesis.

**Specific tumor preferences of RNase IIIa/b hotspot mutants.** The breadth of TCGA and IMPACT sampling across dozens of cancer types allowed us to interrogate potential specificity of DICER1 hotspots. Moreover, while TCGA covers only primary cases, MSK-IMPACT has a strong representation of metastatic samples. Both RNase IIIa and IIIb hotspots, including biallelic

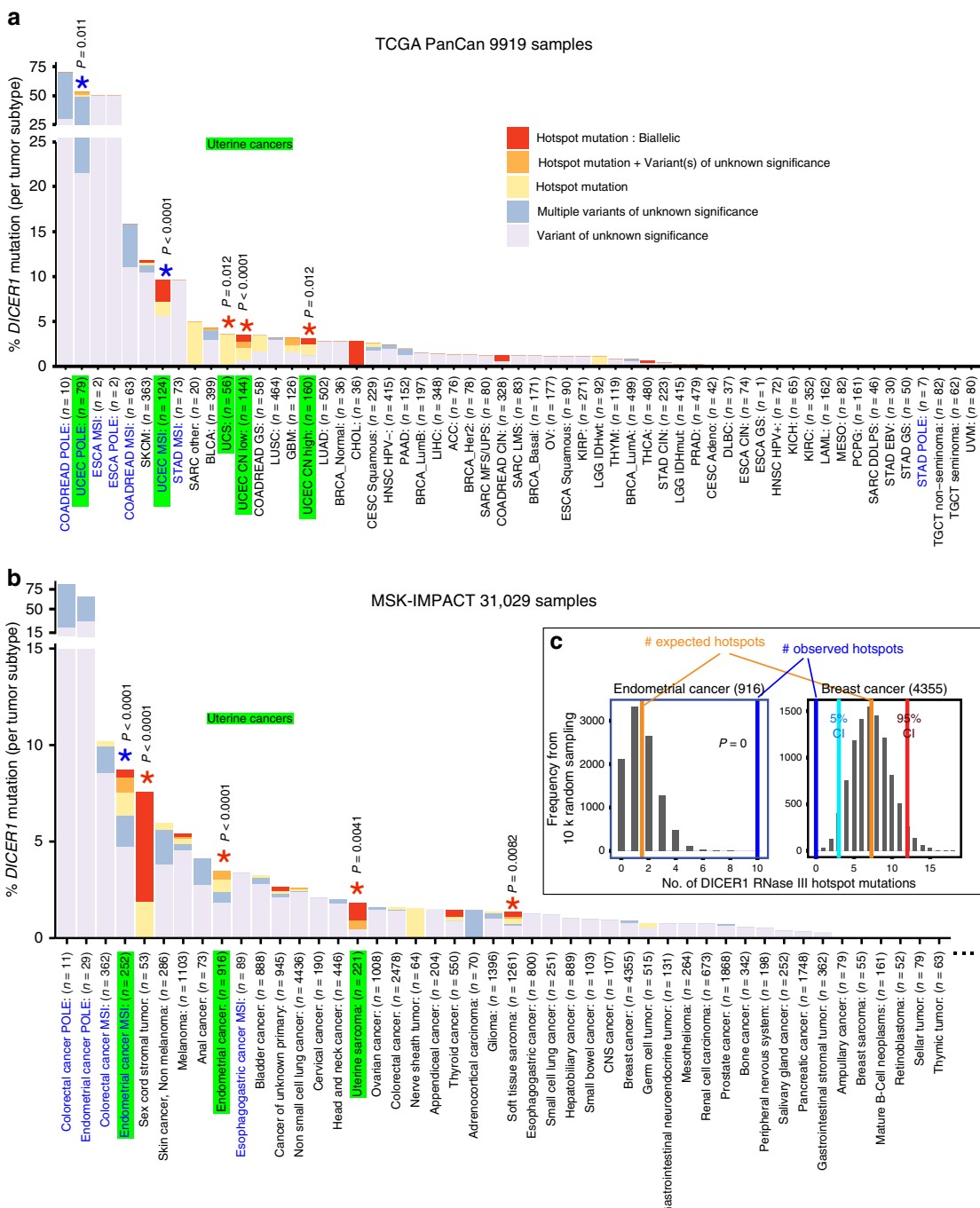

**Fig. 2** Frequency of *DICER1* mutations in TCGA and MSK-IMPACT data across tumor types. We classify alleles as variants of unknown significance, RNase IIIa/b hotspot mutations, and/or biallelic mutations in **a** TCGA-PanCan data and **b** MSK-IMPACT data. Because of the diversity of MSK-IMPACT cohort, only cancer types with >50 cases are shown, except for certain rare MSI/POLE tumors that are shown to match TCGA hypermutated data. Hypermutated tumors are designated in blue. Note the MSK-IMPACT data is generally underpowered relative to TCGA to call copy number loss; thus, only TCGA data includes biallelic calls for HETLOSS. Many commonly surveyed cancer types (*n* for each tumor type summarized below) are devoid of *DICER1* hotspot mutations (also see Supplementary Figs. 2 and 3). However, independently in TCGA and MSK-IMPACT datasets, multiple uterine cancers were enriched for hotspot mutations (green highlighted groups on *x*-axis). Bootstrap resampling analysis with hypermutated cases excluded shows that several uterine cancers and sarcoma subtypes are statistically enriched in both TCGA and MSK-IMPACT datasets (red asterisks). Furthermore, in a second iteration where hypermutated cases were retained in resampling analysis, additional uterine cancers were found enriched for RNase III hotspots (blue asterisks). **c** Examples of statistical analyses for enrichment or depletion of hotspot mutations in MSK-IMPACT cancer groups. From random samplings of 10,000 bootstrap replicates, the average numbers of hotspot mutations obtained are in blue, while observed numbers of hotspot mutations are in orange. For enrichment, endometrial cancer is shown as an example. Statistical significance for depletion can only be estimated using percentile confidence intervals (CI) when sample sizes are large, as illustrated for breast cancer (zero hotspots, falling well below 5% CI). Statistical analyses for all other cancer groups from TCGA and MSK-IMPACT data are shown in Supplementary Figs. 2 and 3

cases, were represented in metastatic tumors (Fig. 1b), suggesting that these mutant cells can support migration, invasion and/or recolonization. Interestingly, the distribution of *DICER1* hotspot mutations was not even across cancer types. For example, visual inspection shows that hotspot mutations and biallelic cases were generally restricted to a few related classes of uterine cancers in both TCGA and MSK-IMPACT datasets (Fig. 2a, b). By contrast, the strong majority of cancer types completely lacked *DICER1* hotspot mutations (Fig. 2b and Supplementary Fig. 3), including some very commonly sequenced cancers (e.g., breast, pancreatic, and prostate cancers, each with >1000 cases).

We performed random resampling analysis (see Methods) to estimate significance for enrichment or depletion of *DICER1* hotspot mutations. This analysis finds uterine cancers are genuinely enriched for hotspot mutations in both TCGA and MSK-IMPACT cohorts. In TCGA data, uterine corpus endometrial carcinoma (UCEC CN high and CN low) and uterine carcinosarcoma (UCS) were statistically enriched for hotspots (Fig. 2a and Supplementary Fig. 2). Similarly, endometrial cancer and uterine sarcoma were significantly enriched for hotspots in MSK-IMPACT (Fig. 2b and Supplementary Fig. 3). The similar enrichments from independent largescale cancer cohorts strongly imply preferred impacts of *DICER1* hotspot alleles in the uterine cancer setting. In addition, soft tissue sarcoma and sex-cord stromal tumor were enriched for hotspots in MSK-IMPACT data (Fig. 2b). This analysis was conservative in that we excluded hypermutated cancers from the resampling procedure. When hypermutated cases were included, we find significant enrichment for hotspots in additional uterine cancer groups (UCEC MSI and UCEC POLE) in TCGA and in endometrial cancer MSI in MSK-IMPACT data, respectively (Fig. 2a, b). However, other hypermutated cancers (POLE or MSI) generally lacked DICER1 hotspots.

Reciprocally, 70% (29/40) of TCGA PanCan and 63% (26/41) of MSK-IMPACT cancer types had zero *DICER1* RNase III hotspot mutations (considering only cancer types with sample size >50). Of these, 8 TCGA and 17 MSK-IMPACT cancer types had sample sizes >200, larger than typical uterine cancer cohorts with clear hotspot enrichment (Supplementary Figs. 2 and 3). This indicates a dearth of *DICER1* hotspot mutations across the vast majority of cancer types. Nevertheless, statistical significance for depletion can only be estimated when sample sizes are sufficiently large that random sampling generates a normal distribution. As an example, the observed hotspot mutations in breast cancer (0/4355 patients) are well below the 5% confidence interval from bootstrap resampling, indicating significant depletion (Fig. 2c). We return to the apparent tumor bias of *DICER1* RNase IIIa/b inactivation in subsequent gene expression analyses.

**Effects of *DICER1* RNase IIIb hotspots on miRNA biogenesis**. The Dicer RNase III domains cleave opposite sides of pre-miRNA hairpins[27], with RNase IIIa cutting the 3p arm and RNase IIIb cutting the 5p arm (Fig. 3a). Cell culture and in vitro processing assays established that cancer hotspot mutations in DICER1 RNase IIIb selectively impair miRNA-5p processing (Fig. 3a). Accordingly, cancer-associated *DICER1* hotspot mutations bias the relative yield of 5p to 3p miRNAs in tumors. While this outcome has been demonstrated in select tumors[24,37], we sought a more comprehensive analysis of miRNA processing asymmetry alterations across cancer small RNA datasets.

We first compared miRNA levels in 15 RNase IIIb hotspot endometrial mutants with 548 other UCEC cases, combining *DICER1* mutants outside of RNase III hotspot sites with all wildtype cases as controls[28,41]. As a baseline, we randomly selected a similar-sized set of non-hotspot endometrial cases

(n = 15) and compared them to the remainder of the control cohort. Although some miRNAs exhibited apparent up/down-regulation in control comparisons, almost none were significant, as expected. More importantly, fluctuating miRNAs showed no directional bias when segregated by 5p and 3p origin (Fig. 3b). In contrast, the 15 RNase IIIb hotspot endometrial cases exhibited directional trends for downregulation of individual 5p strand miRNAs (Supplementary Fig. 4). Note that 5p strand miRNAs were still detected in RNase IIIb hotspot mutants, suggesting impairment but not abrogation of 5' pre-miRNA arm cleavage. We also observed a reciprocal enrichment of 3p strands (Supplementary Fig. 4). While this might imply that RNase IIIb mutations enhance miRNA-3p biogenesis, as libraries were globally normalized to equalize depth, it is conceivable that lost 5p reads were taken up by 3p reads. We address these possibilities later in functional assays. In any case, the expression of 5p and 3p strands in RNase IIIb hotspot mutants were significantly different from controls (Supplementary Fig. 4, Wilcoxon Rank-Sum test; p = 1.2E−34). These data extend previous observations on the asymmetric effects of RNase IIIb hotspot mutants on pre-miRNA hairpin products[25].

We were curious whether RNase IIIb cases with overt biallelic-inactivating mutations were distinct from hotspot samples associated with other mutations of unknown significance, or lacking other DICER1 alterations. It is challenging to definitively ascertain the absence of biallelic mutations in cancer genome sequencing data. Nevertheless, we segregated the four biallelic-inactivated cases from 11 other RNase IIIb hotspot samples, and observed that both groups showed significantly biased outputs in miRNA arm accumulation (Supplementary Fig. 4). However, the biallelic-inactivated cases clearly exhibited more severe bias and were disproportionately responsible for signals detected in the aggregate RNase IIIb mutant analysis (Fig. 3c and Supplementary Fig. 4). Thus, there is a range of functional miRNA pathway alteration among tumors collectively annotated with RNase IIIb hotspot mutations.

We expanded these trends by systematically analyzing 9919 TCGA small RNA datasets[28] encompassing 33 tumor types available from the GDC portal [https://portal.gdc.cancer.gov/]. For this purpose, we wished to move away from expression-based analysis comparing individual miRNAs, which requires matching mutants to control tumor types for any reasonable inference of miRNA up/downregulation. Since few *DICER1* mutants lack multiple cases in individual tumor types, besides endometrial cancer (Fig. 2), this approach could not be utilized broadly. Instead, we summarized the relative abundance of 5p to 3p strands for each patient as $m^i_{53} = \log_2(m^i_5/m^i_3)$, where $m_x$ is the median expression of the $x$-strand miRNAs in the TCGA sample $i$ (Supplementary Fig. 5A). This statistic provides a convenient and simple overview for each sample that is robust amidst the diversity of small RNA expression in different libraries and to potential dominating loci. In particular, as some highly expressed miRNAs in individual libraries show extreme asymmetry according to strand selection rules[42,43], we found the median metric (as opposed to mean) was advantageous in illustrating shifts across heterogeneous and diverse datasets. Supplementary Fig. 5B illustrates how this metric was applied to example DICER1-wt and RNase IIIb hotspot mutant cases, revealing overall shifts in 5p/3p arm distributions across miRNA loci.

As each tissue is characterized by a different spectrum of miRNAs, the typical range of $m^i_{53}$ values varies between tumor types. This was evident when calculating $m^i_{53}$ scores across the entire TCGA dataset, and segregating them by tumor type. For example, the ovarian epithelial tumor cohort (486 cases) exhibited globally lower $m^i_{53}$ metric, although no RNase IIIb hotspots were found in these patients. However, the $m^i_{53}$ metric was otherwise

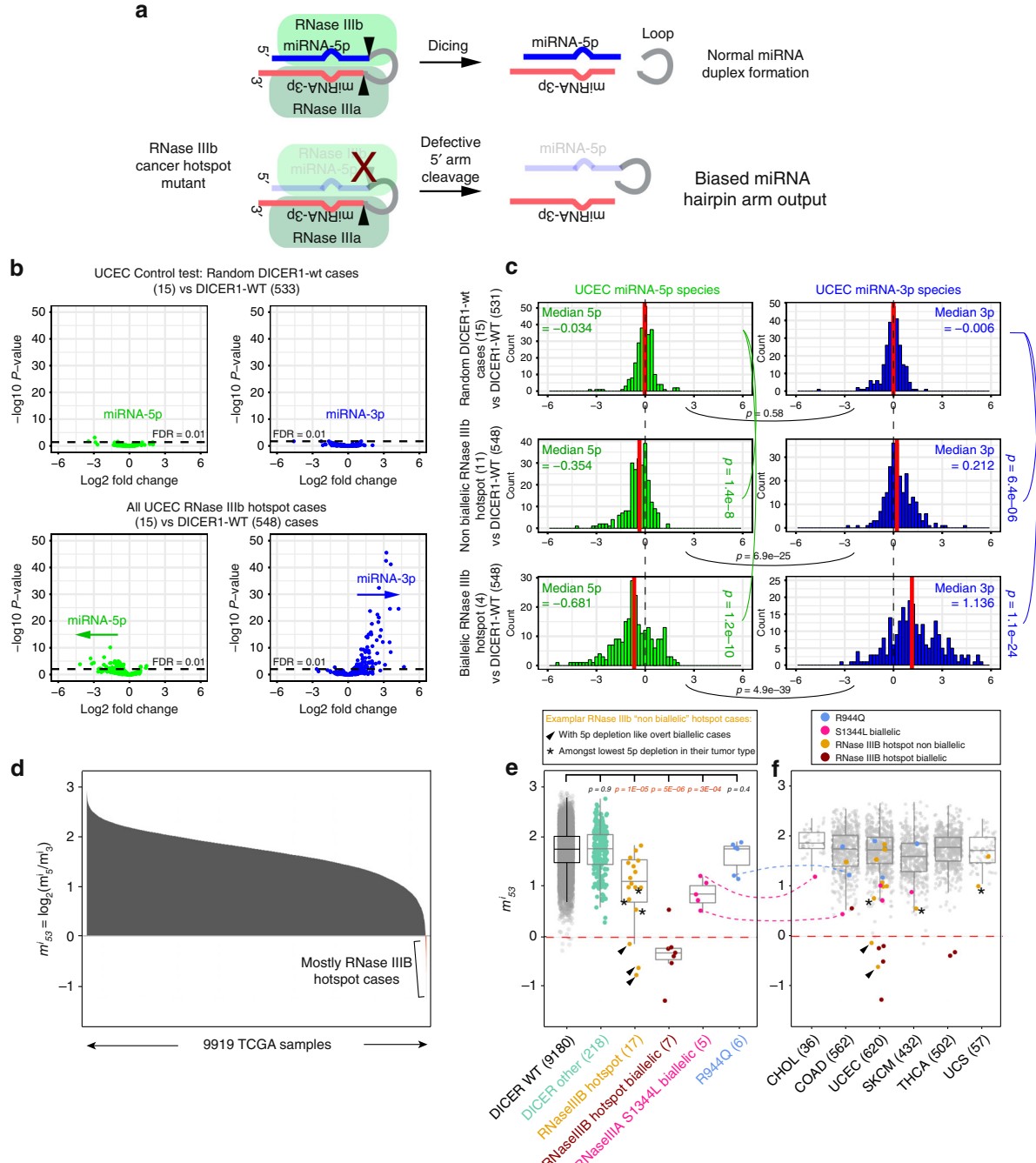

**Fig. 3** Cancer hotspots in DICER1 RNase IIIb and RNase IIIa domains deplete 5p miRNAs. **a** Existing model for how the two RNase III domains of Dicer cleave a pre-miRNA hairpin, and for strand-specific defects in RNase IIIb cancer hotspot mutants. **b** Volcano plots of miRNA-5p (green) and miRNA-3p (blue) expression in sets of Uterine Corpus Endometrial Cancer (UCEC) cases. Top: control comparison of randomly selected DICER1-wt (15 vs. bulk 533 cases) shows only incidental fluctuation of miRNA levels. Bottom: comparison of 15 RNase IIIb hotspot mutant to 548 DICER1-wt UCEC cases shows downregulation of miRNA-5p species and upregulation of miRNA-3p species. **c** Barplots of the same UCEC comparisons showing that RNase IIIb hotspot cases with overt biallelic-inactivating mutations (four datasets) distort miRNA-5p/3p profiles more severely than other RNase IIIb hotspot cases (11 datasets). **d** Systematic analysis of relative miRNA strand processing in TCGA small RNA-seq data. We calculated a metric ($m^i_{53}$) that summarizes sample-specific relative 5p abundances. The few samples with negative values are dominated by RNase IIIb hotspot mutants. **e** Segregation of TCGA samples by genotype. For this analysis, the ovarian serous cystadenocarcinoma cancer (OV) cohort was omitted, due to tissue-specific low $m^i_{53}$ score (Supplementary Fig. 6). *DICER1* mutant samples stratify according to whether they have RNase IIIb hotspot mutants with biallelic-inactivating mutations (in red), or have RNase IIIb hotspots accompanied by other alterations of unknown consequence or lack secondary mutations (in orange). The remainder of *DICER1* mutant samples behave similarly to *DICER1-wt* samples, with the exception of S1344L cases (in green), which also exhibit miRNA-5p depletion. **f** The $m^i_{53}$ metric varies according to the miRNAs expressed in a given tissue. Segregating DICER1-RNase III hotspot cases by tumor type illustrates that biallelic hotspot cases (which include all S1344L cases) typically exhibit lowest $m^i_{53}$ scores within their cohort, whereas other RNase III hotspot cases are more heterogeneous in their behavior. Box plot elements: center line, median; box limits, upper and lower quartiles; whiskers, 1.5x interquartile range

reasonably stable across 32 other tumor types (Supplementary Fig. 6), indicating its value for general comparisons across diverse datasets.

Across 9919 TCGA datasets, ~0.2% exhibited negative $m^i_{53}$ scores, and these were dominated by DICER1 RNase IIIb hotspot cases (Fig. 3d). Strikingly, when excluding ovarian epithelial tumor, 9/12 lowest $m^i_{53}$ scores involved DICER1 RNase IIIb hotspots, emphasizing how distinct their miRNA profiles are across the entire TCGA cohort (Fig. 3e). When we segregated the TCGA by tumor type, we could further see that RNase IIIb hotspot-biallelic loss-of-function cases were usually at the bottom of their respective cancer groups. Besides the UCEC cohort analyzed above, this is seen by the separation of two RNase IIIb biallelic cases in Thyroid carcinoma (THCA) from 478 other cases in this cohort (Fig. 3f) and another case in Colon adenocarcinoma (COAD) (Supplementary Fig. 6). Note also that glioblastoma (GBM) was not profiled by small RNA sequencing, but an earlier microarray dataset exists at the legacy GDAC Firehose portal. These data lack comparable 5p/3p data for the majority of relevant miRNAs, but conducting the analysis with available data revealed a biallelic RNase IIIb hotspot case among the lowest scores in GBM (Supplementary Fig. 6).

Curiously, the remaining RNase IIIb hotspot cases lacking evidence for biallelic-inactivating changes, or bearing other DICER1 somatic changes of unknown consequence, exhibited a much broader range of $m^i_{53}$ scores. Analogous to expression-based analyses conducted with UCEC cases, the $m^i_{53}$ behavior of these non-biallelic RNase IIIb hotspot cases was intermediate to DICER-wt and DICER-RNase III biallelic-inactivating groups (Fig. 3e). However, with this summary metric for each tumor, we could more clearly observe the heterogeneity of individual cases. Some exhibited 5p-miRNA depletion comparable to hotspot-biallelic cases. Potentially, these may harbor undetected, functionally inactivating DICER1 mutations. However, other RNase IIIb hotspot cases had distinctly higher scores. Since each tumor exhibits a characteristic $m^i_{53}$ range, segregating the data by tumor type better illustrated the underlying trends. In this manner, several non-biallelic RNase IIIb hotspot cases still exhibited 5p-miRNA depletion that was substantially low among their respective tumor cohorts (Fig. 3e, f). Besides UCEC, this was also the case in Bladder Urothelial Carcinoma (BLCA), Sarcoma (SARC), Skin Cutaneous Melanoma (SKCM), Uterine Carcinosarcoma (UCS) (Fig. 3f and Supplementary Fig. 6).

Nevertheless, some RNase IIIb hotspot cases exhibited $m^i_{53}$ scores that were typical for their tumor type (Fig. 3e and Supplementary Fig. 6). One possibility is that these samples have lower tumor purity, which may obscure the ability to record functional changes in miRNA processing. This scenario would similarly reduce the available power to detect biallelic changes. However, another possibility is that some of these samples are heterozygous. If so, this might suggest DICER1 hotspot mutations do not act dominantly to bias miRNA biogenesis.

**DICER1 RNase IIIa-S1344L mutants deplete miRNA-5p species.** Since the median $m^i_{53}$ metric had demonstrable utility for largescale analysis of TCGA small RNA data, we investigated whether other DICER1 mutants exhibited characteristic shifts in hairpin arm distribution. Most other DICER1 mutations (i.e., 217 TCGA DICER1 cases bearing only non-hotspot mutations) did not affect miRNA strand asymmetry (Fig. 3e). For example, the six cases bearing the prominent allele R944Q in the PAZ domain were collectively similar to DICER1-wt cases and other DICER1 mutations of unknown consequence (Fig. 3e).

However, in addition to known RNase IIIb hotspot mutants, we observed the five RNase IIIa-S1344L cases (Fig. 1) were

systematically associated with low 5p abundance. Their collective $m^i_{53}$ score range was higher than DICER1-RNase IIIb hotspot cases with clear biallelic-inactivating mutations, and similar to the remainder of RNase IIIb hotspot cases (Fig. 3e). Of note, the S1344L subclass exhibited a tighter distribution than the RNase IIIb hotspots that lacked overt biallelic-inactivating mutations (Fig. 3e). This is consistent with the fact that TCGA S1344L cases were all associated with secondary loss-of-function hits in DICER1, or a double hit in S1344 (Fig. 1b). The functional consequence of S1344L on miRNA arm distribution was even more apparent when separating cases by cancer type. For example, the S1344L case with the highest $m^i_{53}$ score was actually the lowest scoring case in its tumor cohort (CHOL), emphasizing it is truly an outlier (Fig. 3e, f, compare dotted lines connecting individual cases between figures). Similarly, other S1344L cases were among the lowest scores in COAD, SKCM, and UCEC cohorts (Fig. 3e and Supplementary Fig. 6).

Since previous mechanistic studies indicated RNase IIIa and IIIb activities are distinct and can be uncoupled[26] (Fig. 3a), the similar effects of RNase IIIa and IIIb mutants on tumor small RNA profiles was unanticipated. In particular, they suggest that RNase IIIa-S1344L somehow impacts RNase IIIb function during tumorigenesis.

**Functional validation of DICER1 RNase IIIa and IIIb hotspots.** We selected several DICER1 hotspot mutations for functional assays: PAZ domain-R944Q, RNase IIIa-S1344L and RNase IIIb-E1813G (Fig. 1a). We introduced these into a human DICER1 cDNA, and expressed wt and mutant hDICER1 plasmids in Dicer-KO MEFs that we previously used to analyze Dicer-independent miRNA biogenesis[44]. We verified that transfected Dicer proteins accumulate at comparable levels (Fig. 4a). To evaluate potentially selective effects of mutant DICER1 proteins on miRNA-5p vs. miRNA-3p biogenesis, we sought miRNAs that accumulate both duplex strands of a given pre-miRNA, thereby providing internal controls. This was not trivial since there is usually highly asymmetric accumulation of the duplex strands, with one strand (miRNA*) preferentially ejected and therefore typically difficult to detect[3].

We screened a panel of loci based on our previous studies of miRNA* function[45], and identified two miRNAs with suitable properties (mir-151 and mir-199a-1). In particular, their processing was arrested as pre-miRNA hairpins in Dicer-KO MEFs indicating full dependence on Dicer, and both partner strands could be detected when DICER1 was reintroduced into mutant cells. R944Q did not exhibit obvious differences from wild-type DICER1 in this assay. However, both RNase III mutants exhibited specific defects in accumulation of miRNA-5p species, while partner miRNA-3p species were produced at normal levels (Fig. 4b, c). These tests suggested the primary defect in RNase IIIa/b mutants is indeed depletion of miRNA-5p species, not upregulation of miRNA-3p species (Fig. 3b). As further controls, miR-144-3p, Dicer-independent miR-451 and U6 snRNA accumulated similarly in the presence of wt and mutant DICER1 proteins. Thus, DICER1 RNase IIIa-S1344L and RNase IIb-E1813G are selectively compromised in processing canonical miRNA-5p species.

To assess if these biogenesis defects have consequences for miRNA function, we performed luciferase activity assays. We tested a panel of sensor constructs in the presence of different miRNA combinations of miRNA and Dicer constructs. As expected, repression capacities of mir-151 and mir-199a-1 on their cognate 5p and 3p sensors was rescued by wild-type DICER1. Furthermore, R944Q provided similar rescues, indicating that it is not functionally compromised in this context. By

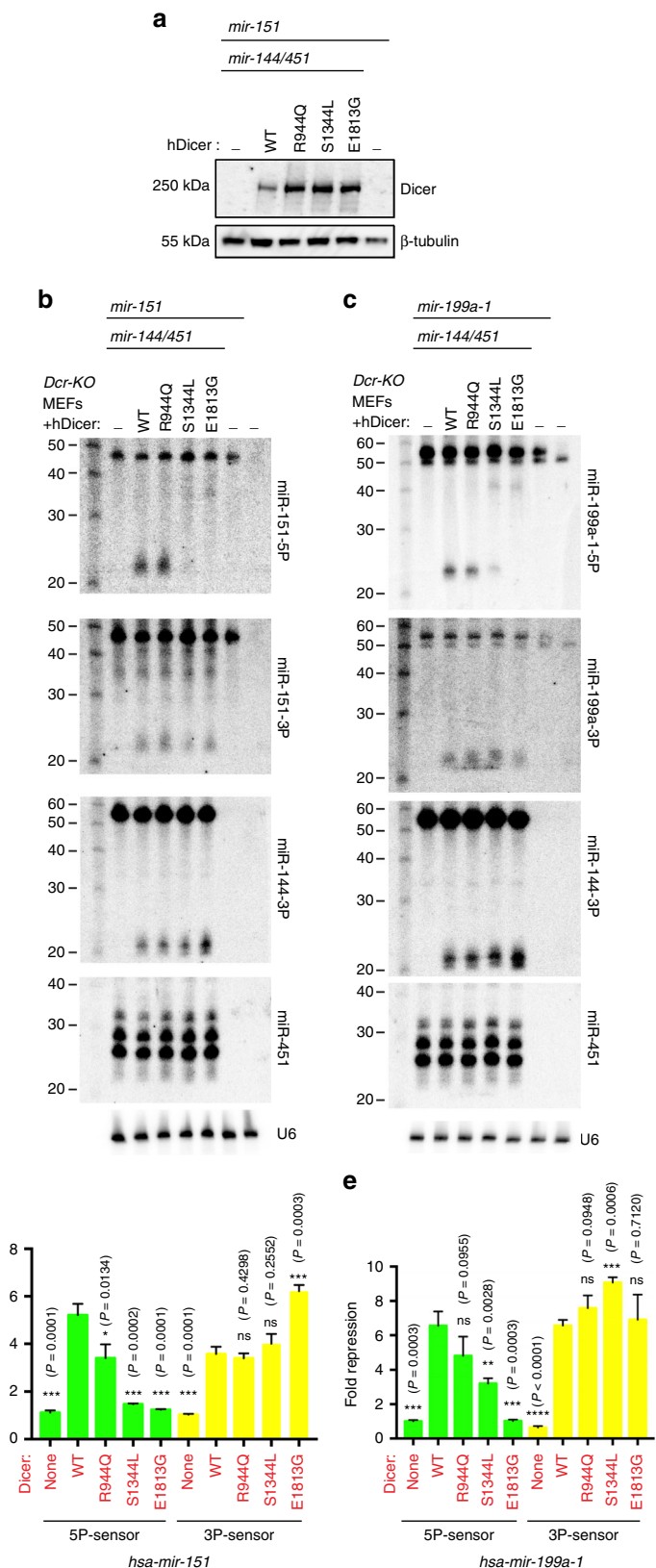

contrast, S1344L and E1813G rescued miRNA-3p activity, but were impaired in conferring miRNA-5p activity for both miRNAs (Fig. 4d, e). Of note, while *mir-151* and *mir-199a-1* had overall similar behavior across this panel of DICER1 mutants, miR-199a-

1-5p was slightly matured by S1344L compared to E1813G (Fig. 4d) and correspondingly had measurable activity on its sensor in S1344L but not in E1813G cells (Fig. 4e). This is consistent with the globally lower miRNA-5p depletion observed

**Fig. 4** *DICER1* RNase IIIa/IIIb hotspots selectively affect miRNA-5p processing and activity. **a** Western blot validation of accumulation of wt and mutant DICER1 proteins in *Dcr-KO* MEFs. Cells were transfected with the indicated Dicer and miRNA constructs blotted using human Dicer antibody and ß-tubulin as loading control. **b**, **c** Northern blot assays of small RNAs from *Dcr-KO* MEFs transfected with the indicated Dicer and miRNA constructs. The top two blots are for 5p and 3p probes directed against *mir-151* (**b**) and *mir-199a-1* (**c**), which detect different mature 21–23 nt species but co-detect their cognate pre-miRNA hairpins. The lack of mature species without Dicer transfection confirms their full Dicer dependence. Both miRNAs yield relatively similar 3p species with wt and mutant Dicer proteins, while 5p species are selectively impaired. mir-144-3p serves as another canonical miRNA control, miR-451 is a Dicer-independent miRNA, and U6 is a loading control. Note that some structured pre-miRNA hairpins run faster than their predicted linear sizes; *pre-mir-151* is expected to be 58 nt, *pre-mir-151* is expected to be 63 nt, *pre-mir-144* is 57 nt. **d**, **e** Luciferase sensor assays. *Dcr-KO* MEFs were transfected with the indicated human DICER1, miRNA and sensor constructs. With reference to the lack of repression of these miRNAs on their sensors in *Dcr-KO* MEFs, wt and mutant DICER1 proteins were able to rescue 5p/3p activity of miRNAs with the exception of strongly diminished or lack of 5p rescue by RNase IIIa/b mutant DICER1 proteins. Standard error of triplicate luciferase sensor experiments is shown

in RNase IIIb biallelic cases compared to S1344L biallelic cases (Fig. 3d), even though both classes comprise strong outliers across TCGA data.

Although these assays were limited, PAZ-R944Q lacked substantial impact. We note that while arginine is only the ninth-most common amino acid, it is the most frequently altered amino acid in cancer[46]. This may be a consequence of the high frequency of CG in arginine codons (4 out of 6), coupled with the high mutagenicity of CG dinucleotides. Thus, PAZ-R944Q might be incidental. On the other hand, these experimental assays validate the unexpected finding that cancer-associated hotspot mutations in DICER1 RNase IIIa and RNase IIIb domains actually have similar biochemical and function consequences.

**RNase IIIa-S1344 is coupled to the RNase IIIb active site**. As S1344 is located in the RNase IIIa domain, we sought mechanistic insight as to why its mutation leads to 5p, rather than 3p, depletion. Since intra-molecular dimerization of the two RNase III domains are required for DICER1 function[47], we explored whether structural and conservation statistical analysis could explain the mutation effect.

Structural information on human DICER1 RNase III domains has long been limited, and until recently, only a homodimeric crystal structure of its RNase IIIb domain (2eb1) [https://www.rcsb.org/structure/2eb1] was available[48]. Alignment of human DICER1 RNase III domains shows S1344 is homologous to T1733 in RNase IIIb (Supplementary Fig. 7A). We modeled S1344 into a model of the RNase IIIa-IIIb heterodimer, as inferred from the RNase IIIb homodimer structure[48]. In the heterodimer model, S1344L (in RNase IIIa) and its homologous residue T1733 (in RNase IIIb) are far from the active site residues (19.60 ± 2.62 Å distance) in their respective domains. Instead, S1344L is closer (11.7 ± 2.0 Å distance) to the active site of domain IIIb (residues E1813, D1810, D1709, E1705) and T1733 in is close to the active site residues of RNase IIIa (Supplementary Fig. 7B). This suggests how S1344L may disrupt RNase IIIb activity.

Very recently, cryo-EM structures of full-length DICER1 were reported[49], allowing us to examine inter-domain contacts of RNase IIIa/b in their native conformations. Consistent with heterodimer modeling, we observe juxtaposition of the RNase III domains, with RNase IIIa-S1344 on the heterodimer interface with RNase IIIb in all three available structures (5zak [https://www.rcsb.org/structure/5zak], 5zam [https://www.rcsb.org/structure/5zam] and 5zal [https://www.rcsb.org/structure/5zal]), only 3–4 Å from F1706. Since F1706 neighbors catalytic residue E1705 that binds the active site $Mg^{2+}$ (which we modeled into the cryo-EM structure based on 2eb1 [https://www.rcsb.org/structure/2eb1] homodimer structure), this brings S1344 in proximity to the RNase IIIb active site (~8–9 Å from E1705, Fig. 5a). While the published DICER1 structures are not in their catalytic state, nor have high enough resolution for $Mg^{2+}$ ions to

be positioned precisely[49], it is conceivable that the S1344 side chain hydroxyl group is even closer to the active site in an, as yet, unseen active conformation. This could involve the hydroxyl group of S1344 connecting through water or directly to $Mg^{2+}$.

An independent strategy to evaluate the importance of S1344 is to explore its constraint via interactions with active site residues of the sister RNase III domain across the protein's evolutionary history. Here, we exploited evolutionary couplings (EC) methodology, which uses a multiple sequence alignment built from thousands of homologous sequences along with a statistical maximum entropy framework to identify pairs of residues that are evolutionarily constrained as interacting pairs[50]. Evolutionary couplings inferred only from co-variation patterns in sequences, without reference to known three-dimensional (3D) structures, were successful in ab initio prediction of correctly folded structures[50,51] and in identifying alternative conformations[52,53]. The top-ranked pairs of evolutionarily coupled residues (ECs)—about $L$ of all possible $L^2$ pairs where $L$ is the length of the protein sequence—are typically enriched in residues directly critical for structure and/or function[51].

We subjected human DICER1 1271–1829 comprising the two RNase III domains to evolutionary coupling analysis. For the well-structured DICER1 RNase IIIb domain, 70% of paired residues in the top 100 evolutionary couplings (ECs) were within 5 Å of each other in the 3D structure (Fig. 5b), providing evidence that evolutionary couplings analysis can identify constrained interacting residue pairs from sequence information alone. Reassuringly, intra-domain couplings among each of the four RNase IIIa active site residues, as well as the four RNase IIIb active site residues, were recovered among the top couplings, indicative of their known functional coordination. In particular, RNase IIIb E1705-D1709 active site residues comprised the 9th highest ranked coupling, and six couplings between RNase IIIb active site residues and four couplings between RNase IIIa active site resides were detected in the top 140 ECs (Supplementary Data 4).

Our previous studies revealed numerous examples in which potentially anomalous signals within a monomer actually derive from a homodimeric interaction[50]. Consistent with this interpretation, we observe lower among the top ECs, but still within the top $L$ hits, examples of anomalous inter-domain RNase IIIa-b active site couplings (Supplementary Data 4). This provides context for the bona fide, strong, inter-domain ECs that align precisely with the RNase IIIa/b interface shown in the cryo-EM structures (e.g., 5zak [https://www.rcsb.org/structure/5zak], Fig. 5b). Notably, we observe coupling of T1733 (the cognate of S1344) to active site residue RNase IIIb-E1705 in the top $L$ hits, as well as direct coupling of RNase IIIa-S1344 in RNase IIIb-F1706 in the top 1.5% of all detected couplings (Supplementary Data 4). This is consistent with close physical proximity of these residues in the cryo-EM structure (Fig. 5a). Thus, ECs inferred from

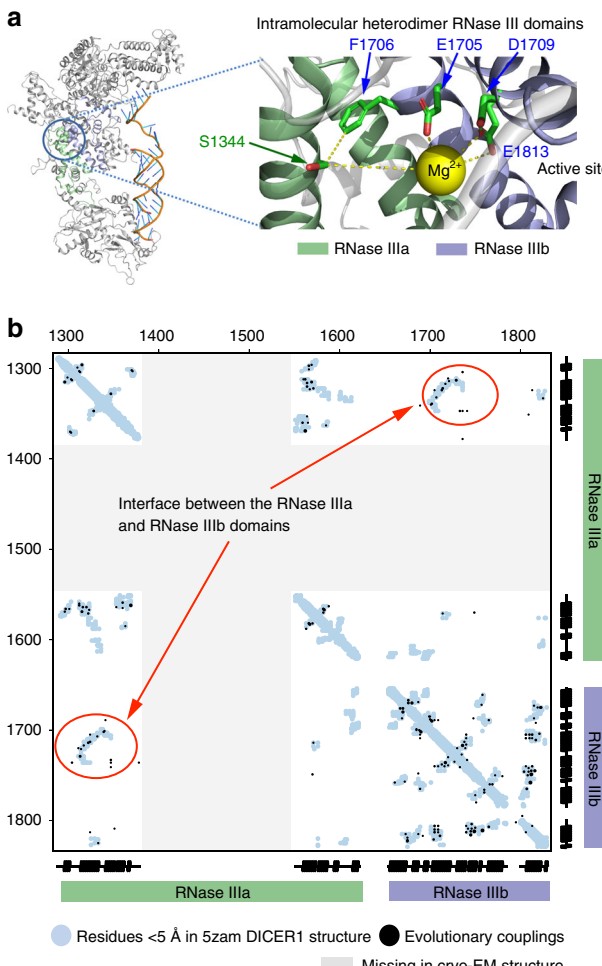

**Table 1 Significantly deregulated genes in *DICER1* hotspot endometrial cancer**

| Genes | logFC | adj *p*-value |
|---|---|---|
| *HMGA2* | 3.708 | 1.662E-6 |
| *IGDCC3* | 3.648 | 4.091E-5 |
| *ACVR2B* | 1.211 | 1.365E-4 |
| *MMP16* | 2.333 | 4.123E-3 |
| *C17orf63* | 0.782 | 4.577E-3 |
| *ADAMTS7* | 1.993 | 1.247E-2 |
| *IGF2BP2* | 3.294 | 2.501E-2 |
| *FAM171B* | 1.801 | 3.499E-2 |
| *MGAT5B* | 2.875 | 3.851E-2 |

A differential gene expression analysis comparing *DICER1* hotspot mutants to wildtypes. We compared the gene expression levels in eight *DICER1* mutants to the levels in 222 *DICER1* wildtypes using the *limma voom* toolkit. We used Bonferroni correction to adjust our *p*-values for multiple hypothesis testing ($p_{adj}$ < 0.05). logFC: change in gene expression (log based). All genes significantly changed in mutants were upregulated, none were significantly downregulated

**Fig. 5** Structural rationale for how RNase IIIa-S1344 affects RNase IIIb function. **a** (Left) The human DICER1 cryo-EM structure 5zam [https://www.rcsb.org/structure/5zam] is shown, and part of the RNase IIIa domain (in green)/RNase IIIb (in purple) interface is enlarged. Inspection of this region reveals that RNase IIIa-S1344 resides on the inter-molecular heterodimeric interface with the RNase IIIb domain, closest to F1706, which is adjacent to the active site residue E1705. E1705, D1709, and E1813 coordinate the Mg2+ ion in higher resolution structures of the RNase IIIb domain (i.e., 2eb1) and the Mg2+ ion is modeled here by overlaying 2eb1 [https://www.rcsb.org/structure/2EB1] with 5zam [https://www.rcsb.org/structure/5zam]. This places S1344 within 8–9 Å of the RNase IIIb domain active site Mg2+. Modeling the side chain of S1344 and correct placement of the Mg2+ in the catalytic state may bring the hydroxyl of the serine even closer. **b** Contact map summarizing highly evolutionarily coupled residues, analyzed across the equivalent of DICER1 aa1271-1829, comprising the RNase IIIa/b domains. Highly coupled residue pairs (black) were displayed on top of residues located <5 Å apart in the 5zam [https://www.rcsb.org/structure/5zam] cryo-EM structure (aqua). Note that the RNase IIIa domain contains a large flexible insertion whose structure is unknown (gray). This analysis reveals not only functionally coupled residues within each RNase III domain, as expected, but also a prominent interface of RNase IIIa/IIIb interactions that includes S1344. These observations provide an evolutionary explanation for how cancer hotspot RNase IIIa-S1344L mutations impair RNase IIIb activity

sequence information alone, reinforce the notion of an evolutionarily constrained, functionally relevant inter-domain interaction between S1344 and the RNase IIIb catalytic center that is required for proper pre-miRNA processing.

**Specific derepressed gene sets in DICER1 hotspot cancer.** Having expanded the set of functional mutations in human DICER1, we investigated whether these lead to interpretable changes in gene expression. Others modeled effects of DICER1 hotspots on gene expression in mouse *Dicer*-mutant cell lines[24], but there have been limited attempts to connect differential miRNA expression in *DICER1* hotspot mutants with mRNA changes in human tumors. Since *DICER1* hotspot mutations are overall rare in cancer, we focused on UCEC[54], which contained the largest number of *DICER1* hotspot mutants (Figs. 1 and 2). Analyzing UCEC RNA-Seq data, we note nine genes significantly upregulated, and none downregulated, in RNase IIIa/b hotspot cases compared to other samples (*p* < 0.05 after Bonferroni correction; Table 1). Derepressed genes include *HMGA2*, an oncogene target of let-7 family members[55–57], and other oncogenes such as *IGFBP2*[58,59] and *MMP16*[60].

We next asked whether upregulated genes in *DICER1* hotspot mutants were enriched for targets of particular miRNA families. We conducted gene-set enrichment analysis (GSEA) using well-known biological pathways and well-conserved miRNA family target genes as our query gene sets[61]. Most (5 out of 7) enriched gene sets represented miRNA family targets (Fig. 6a), suggesting that gene expression signatures in RNase IIIa/b hotspot mutants are dominated by certain depleted miRNA families rather than common biological pathways.

The strongest enrichments were seen for upregulation of genes bearing target sites for let-7 or miR-17 seed families (Fig. 6a, b; FDR < 10%). For both families, the 5p-miRNA is the predominant strand and as expected, 5p-strand miRNAs of these families were downregulated relative to 3p-strands in RNase IIIa/b mutants (Fig. 6c). Another 5p-depleted family with target upregulation was miR-15/16 (Fig. 6a; FDR = 11%). Of note, let-7[55–57] and miR-15/16[62] are known tumor suppressor miRNA families. Interestingly, for the other two families, miR-29 and miR-101 (Fig. 6a; FDR = 11%), although their predominant miRNA strands are 3p, both mature and passenger strands of these families were downregulated in *DICER1* RNase IIIa/b mutants (Fig. 6c). This is consistent with observed derepression of their respective target cohorts, but suggests that reduced expression of these miRNA families is an indirect effect of *DICER1* hotspot mutations. Finally, GSEA analysis indicated modest enrichment for NOTCH-related pathways (Fig. 6a, b; FDR<15%), which is well-appreciated to be mutated in cancer[63].

Overall, these analyses indicate that alterations in pre-miRNA processing in *DICER1* biallelic endometrial tumors yield preferential effects on derepression of cognate direct targets of

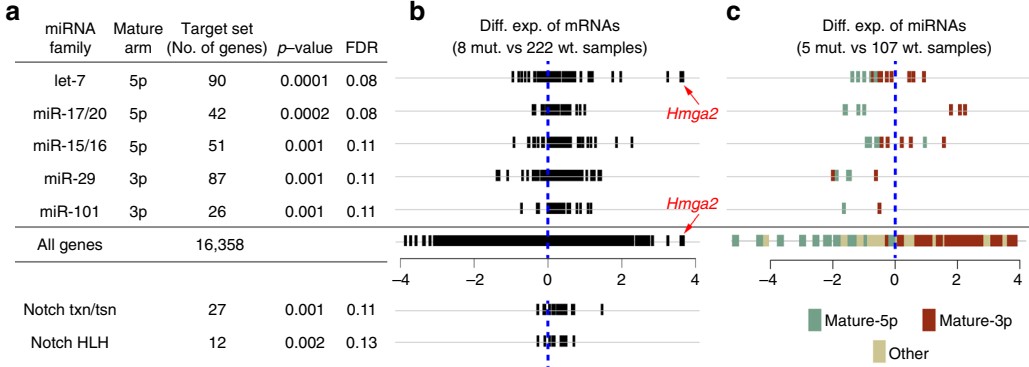

**Fig. 6** Gene expression signatures of DICER1 hotspot UCEC tumors. **a** Gene-set enrichment analysis (GSEA) on mRNA profiles of DICER1 RNase III hotspot cases in uterine corpus endometrial cancer (UCEC). Shown are all enriched sets in RNase III hotspot samples; five are miRNA target sets and two relate to Notch pathway targets. **b** The expression distributions of all enriched gene sets were preferentially upregulated in DICER1 mutants compared to wildtypes. HMGA2, a well-known oncogene and let-7 target, was the most-deregulated gene. **c** For each of the three mature-5p miRNA families, we saw consistent downregulation of 5p strand (green) and upregulation of 3p strand (red) miRNA members in DICER1 hotspot mutants. For the mature-3p miRNA families, both arms were downregulated in mutants. mut: DICER1 hotspot mutant; wt: DICER1 wildtype; Diff. Exp.: Differential expression (log$_2$ ratio of mRNA/miRNA levels); p-value: The probability for the null hypothesis that the genes in the set are not differentially upregulated in mutants compared to wildtypes; FDR: p-value corrected for multiple hypothesis testing

specific miRNAs, and certain likely indirect miRNA target networks and associated gene expression signatures that may be relevant for DICER1 hotspot uterine cancers.

## Discussion

We systematically analyzed the landscape of DICER1 hotspot mutations that lead to miRNA strand asymmetries in cancer. This revealed cancer-subtype enrichments of known RNase IIIb hotspot mutations, which impair processing of miRNA-5p strands, and stratify the effects of tumors with biallelic inactivation on miRNA processing. Importantly, we also functionalize a recurrent biallelic DICER1 mutation in RNase IIIa (S1344L) that causes similar 5p depletion. This was unexpected given that the two Dicer RNase III domains independently cleave opposite arms of the pre-miRNA hairpin.

Interestingly, our cross-cancer analyses reveal DICER1 RNase IIIa/b hotspot mutations in restricted tumor types (Fig. 2). Although DICER1 hotspots are overall rare, and occur sporadically in diverse cancer classes[25] (Supplementary Data 1 and 2), we consistently observe their strongest enrichment in endometrial cancers and in both TCGA and MSK-IMPACT cohorts. Although DICER1 RNase IIIb mutations were previously studied in endometrial cancer[24], we broadly extend this phenomenon and show these events are depleted in many individual cancers, and in cancers in aggregate. This implies there are selective advantages to DICER1 hotspot cancer cells in only certain tissue settings.

In accordance with this notion, we provide evidence for preferential miRNA target upregulation in DICER1 RNase IIIa/b hotspot tumors (Fig. 6). Although we are circumspect to interpret gene expression signatures from the limited number of cases, it seems notable that while many miRNA-5p species are reduced (although not eliminated, Fig. 3), we detected significant changes for target cohorts of only a few specific miRNA families (Fig. 6). These few include known tumor suppressors such as the let-7 and miR-15/16 families, both of which comprise 5p mature miRNA species whose defective biogenesis is a direct consequence of biallelic RNase IIIa/b hotspot mutations in DICER1.

The let-7 family has previously been suggested as a tumor suppressor miRNA family that could be compromised by DICER1 RNase IIIb cancer hotspot mutations[24,26,64], but clear linkage to endogenous DICER1 tumor signature has not been established.

We observed that derepression of let-7 targets was the strongest GSEA signature obtained in DICER1 endometrial cancer, with known let-7 oncogene targets HMGA2 and IGFBP2 strongly deregulated (Table 1). In addition, while miR-15/16 is also recognized as tumor suppressor family, its impact has mostly been restricted to liquid cancers, in particular chronic lymphocytic leukemia[62]. Nevertheless, these miRNAs are expressed outside of the hematopoietic system, and our data potentially expand the functional impact of miR-15/16 to certain solid cancers, such as endometrial cancer. We identify a number of miRNA targets derepressed in DICER1 hotspot tumors (Table 1) as candidates for evaluating potential involvement in oncogenesis.

Overall, these integrated analysis of cancer genome, small RNA, and transcriptome profiling reveal tissue-specific accumulation of DICER1 RNase III hotspot mutations, including an allele within the RNase IIIa domain that affects RNase IIIb function. Thus, cancer genetics reveals unexpected features of Dicer enzymology, pre-miRNA processing, and expression signatures in tumors.

## Methods

**Identification of DICER1 hotspot mutants and biallelic cases.** We used OncoKB (oncokb.org) and Cancer Hotspots (cancerhotspots.org) to identify recurrent hotspot mutations in DICER1. These included well-characterized alterations of RNase IIIb metal-binding residues E1813, D1810, D1709, E1705[25], and the newly characterized allele in RNase IIIa S1344L. We added G1809, which was reported in the literature[31] as a hotspot, as it was recurrent in MSK-IMPACT data. We then identified DICER1 mutations using the TCGA MC3 2+ callers mutation data[29]. Of 9919 samples in the TCGA PanCan dataset, 217 bore somatic DICER1 mutations, of which 31 harbored RNase IIIa/b hotspot mutations (one patient with 2). Similarly, analysis of 31,029 MSK-IMPACT patients with annotated clinical data and tumor subtypes yielded 597 somatic DICER1 mutant cases, of which 57 cases contained RNase IIIa/b hotspot mutations. To identify likely biallelic cases affecting DICER1, we searched for hotspot alleles that co-occurred with another hotspot mutation, truncating mutation; we also searched for HETLOSS but only in TCGA data, since MSK-IMPACT is generally underpowered to call copy number loss.

**miRNA-seq data.** TCGA miRNA sequencing data was downloaded from GDC data portal at the National Cancer Institute [https://portal.gdc.cancer.gov/]. Premapped miRNA sequencing files in bam format were downloaded using the gdc-client tool provided in the GDC data transfer tool package (https://github.com/NCI-GDC/gdc-client), using specific case IDs from TCGA. We obtained a total of 10725 bam files, which included data from both cancer and normal tissues that can be easily distinguished using TCGA sample type identifier ID. Data mapping quality and mapping statistics for the TCGA dataset was obtained using the bam_stats.py tool from the RSeQC package (http://rseqc.sourceforge.net/).

Mappability ranged from 74–99% across TCGA datasets with an average mapping rate of 96.1%. Hence, we retained all datasets for further analysis. Using sample tags to differentiate cancer versus normal tissues, we excluded 632 samples that were derived from normal tissues. In addition, 139 samples with duplicate IDs originating from cancer tissues were excluded owing to uncertain nature of duplicate profiling of these samples (Supplementary Data 3). In sum, we analyzed a total of 9919 TCGA miRNA-seq datasets in this study.

**miRNA set analyzed across TCGA cancer samples**. We obtained 5p and 3p annotations for 992 miRNAs from miRBase (www.mirbase.org), and supplemented this with 210 miRNAs with defined 5p and 3p from in-house miRNA annotations from aggregate small RNA mapping. For broadly conserved miRNA families, we used a list of 176 pan-mammalian miRNAs[65]. Next, we examined the expression of all these miRNAs across TCGA samples and identified 227 broadly expressed miRNAs have average $\log_2$ expression ≥5 across all pan-cancer samples. We took the union of the broadly expressed miRNAs and the broadly conserved mammalian miRNAs to arrive at 280 miRNAs used for subsequent analyses (Supplementary Data 5). Note that in some cases, a given miRNA family is represented by multiple loci with identical 5p and 3p species, and we designated those as single loci (e.g., *hsa-mir-124-1, -2, -3* are all identical on both 5p and 3p arms).

**miRNA counts**. We obtained miRNA expression counts for TCGA PanCancer datasets using the featureCounts software from the Subread package[66]. To obtain counts, we used annotated miRNA-5p and miRNA-3p coordinates flanked with additional two nucleotide sequence on either side, thereby including four additional nucleotides for every miRNA-arm. The counts were normalized using the DEseq package in R, for comparison across the TCGA cancer samples[67]. miRNA-Seq data for Glioblastoma Multiforme cancer study was not available from GDC, therefore, for GBM, TCGA Level 1 microarray expression data [http://firebrowse.org/?cohort=GBM] were processed and normalized using the *AgiMicroRna* R package.

**Analysis of miRNA-seq data**. To identify shifts in the expression of individual miRNA-5p and miRNA-3p species, we utilized the UCEC cohort, which had sufficient numbers of RNase III hotspot mutants for statistical analyses. We performed miRNA-arm differential expression analysis with these sets: (1) all RNase IIIb hotspot mutations (15 cases), (2) RNase IIIB non-biallelic mutations (11 cases), (3) RNase IIIb biallelic mutations (four cases), versus all other samples as controls (bearing non-hotspot DICER mutations or DICER-wt alleles, 548 cases). We also performed a control analysis by randomly selecting 15 UCEC samples lacking hotspot mutations, and comparing them to the remaining samples. Differential expression was performed using the DEseq package in R, and p-values for fold-change analysis were adjusted using Benjamini–Hochberg correction. We defined significant fold-change between miRNA-5p and miRNA-3p species at a False Discovery Rate (FDR) of 0.01.

**Median 5p/3p ratio calculation**. To ascertain bias in miRNA-5p/3p ratio, we used a scoring metric to evaluate the relative production of miRNA-5p and -3p strand across TCGA libraries. Briefly, for every TCGA sample, the miRNA counts from miRNA-5p and -3p arm counts were ranked from lowest to highest to obtain a distribution, from which a median $\log_2$ miRNA expression was estimated for 5p, and 3p arms separately in our restricted set of 280 miRNAs (Supplementary Fig. 5). As calculating mean as a metric for central tendency can be biased by highly expressed loci, we instead chose a median metric, which is more robust when samples across different cancer tissues are compared. Then, a median 5p/3p ratio was calculated using the formula $m^i_{53} = \log_2(m^i_5/m^i_3)$, where $m_x$ is the median expression of the x-strand miRNAs in the TCGA sample $i$. A shift in the median 5p/3p ratio between WT and RNase IIIb hotspot mutations in different cancer types is interpreted as a ratio bias, which can be attributed to compromised processing of miRNA-5p and -3p arms.

**Statistical analysis of DICER1 hotspots across cancer types**. To determine whether certain types of tumors are enriched for *DICER1* hotspot mutations in TCGA and MSK-IMPACT datasets, we performed a resampling test where we generated a distribution from random sampling for each tumor type, and assessed significance for enrichment via this bootstrap procedure. We performed $n = 10,000$ bootstrap replicates, and estimated p-value for enrichment of hotspot mutations by calculating the ratio for number of events where hotspot mutations ($m$) from resampling was greater than the observed number of hotspot mutations to $n$ bootstrap replicates ($m >$ observed/10,000). We performed this analysis in two iterations. In the first, to be conservative, we excluded the hypermutated cancer subtypes, as these cases may falsely influence the occurrence of hotspot mutations during a random sampling procedure. In a second iteration, we retained the hypermutated cases in the random sampling analysis. p-values were then corrected for multiple tests using the Bonferroni correction method.

DICER1 RNase III hotspot mutations are overall rare in both TCGA and MSK-IMPACT datasets (31/9919 samples for TCGA and 57/31029 samples for MSK-

IMPACT. Nevertheless, estimating statistical significance using a bootstrap resampling method is only reliable for enrichment, but not for depletion of hotspot mutations, unless the sample sizes are very large. 13 TCGA and 33 MSK-IMPACT cancer subtypes have sample sizes less than 50, which are too narrow to compute statistical significance. Therefore, we focused only on 40 TCGA and 41 MSK-IMPACT cancer types with sample size >50 for bootstrap resampling analysis (Supplementary Figs. 2 and 3). In cases where random sampling generates a normal distribution, the percentile confidence intervals for the variance statistic can be computed. Using this approach, we were able to reliably estimate statistical significance for depletion of hotspot mutations for two cancer types in the MSK-IMPACT data. Still, we note >20 individual cancer types with >200 cases that completely lack DICER1 RNase III hotspot mutations (Supplementary Figs. 2 and 3), cohorts that are generally much larger than all of the endometrial/uterine cancers that reliably accumulate multiple RNase III hotspots, and ~70% cancer subtypes completely lack hotspot mutations. Thus, there appears to be a general depletion of DICER1 RNase III hotspot mutations across many cancers.

**Identifying evolutionary couplings in RNase III domains**. Our analyses of small RNA data and DICER mutant constructs revealed that RNase IIIa-S1344L variant unexpectedly compromised the biogenesis of 5p-strand miRNAs, which are cleaved from pre-miRNA by DICER1 RNase IIIb. Based on the fact that RNase III dimerization is necessary for proper DICER1 functioning, we wanted to see how S1344L could affect 5p miRNA processing. For this we ran evolutionary couplings (ECs) analysis using code provided at [https://github.com/debbiemarkslab/EVcouplings]. We used residues 1271–1829 of human DICER1 [https://www.uniprot.org/uniprot/Q9UPY3] as the input sequence containing both RNase IIIa and RNase IIIb domains. Our chosen alignment had 36,169 sequences with 69.7% of residue columns containing less than 30% gaps (alignment produced from a normalized bitscore cutoff of 0.3 from the Uniref100 January 2019 release). Pseudolikehood maximization (PLM) was used to infer the ECs. The full evolutionary couplings data is provided in Supplementary Data 4.

**Analysis of RNA-seq data**. To test whether *DICER1* hotspot mutants had distinct gene expression profiles compared to other samples, we obtained processed and normalized RNA-seq datasets (Level 4) from TCGA analysis runs as generated with the Firehose analysis pipeline [https://gdac.broadinstitute.org/]. Most TCGA tumor cohorts had <3 RNA-seq datasets for RNase IIIa/b hotspot mutants, hindering a statistically robust comparison. Therefore, we restricted this analysis to the UCEC study, utilizing there were eight *DICER1* RNase IIIa/b hotspot mutant and 222 *DICER1* wildtype and non-hotspot samples with RNA-seq data. We conducted differential gene expression analysis using the *limma voom* R package[68] on the gene-level RSEM counts.

**Gene-set enrichment analysis (GSEA)**. To test if genes upregulated in UCEC *DICER1* hotspot cases compared to wildtypes were targets of particular miRNAs or members of canonical pathways, we utilized GSEA. To create gene sets for targets of the well-conserved miRNA families, we first downloaded predicted miRNA targets from TargetScan (Release 6.2) and aggregated these using miRNA family-member associations to obtain a list of targets for each miRNA family[5]. We next filtered out predictions with conservation score lower than 90% and collected targets that were in the upper 5th percentile considering their context score (i.e., scores lower than −0.3555). Using these filtered predictions, we created gene sets that were compatible with the conventional GSEA analysis[61].

We combined these miRNA target gene sets with gene sets representing well-known and curated Reactome pathways from MSigDB[69,70]. This gave us a total of 719 gene sets, consisting of 674 gene sets for pathways and 45 for targets of miRNA families. For the GSEA, we utilized the *romer* utility from the *limma* toolkit and used the contrast model that we used in the RNA-Seq data analysis[71]. We set the number of rotations to 10,000 and for each gene set, tested whether the genes in the set were enriched for any direction (up- or downregulation).

We found genes in seven different sets to be significantly enriched towards upregulation and none in the reverse direction (FDR < 0.15; Table 1). Five out of 7 gene sets were representing target genes for miRNA families and three of these were miRNA families for which 5p strand was the predominant strand according to miRBase[72].

**Construction of *DICER1* mutants**. Mutant *DICER1* constructs (R944Q, S1344L, E1813G) were made through site-directed mutagenesis protocol (Agilent Quik-Change) of mammalian expression construct of Flag-HA-tagged human *DICER1*, Addgene plasmid #19881[73]. Sequencing of clones confirmed presence of desired mutations and absence of off-target mutations in rest of the gene. Primers used for site-directed mutagenesis:

R944Q
5′-GTACACATCAGCTACATAAAATTGATGAGGCTGATCAAAATTGCG-3′
5′-CGCAATTTTGATCAGCCTCATCAATTTTATGTAGCTGATGTGTAC-3′
S1344L
5′-TTTTTGCTTCTCATATATAAAAGGCGGCCCTCATGCG-3′
5′-CGCATGAGGGCCGCCTTTTATATATGAGAAGCAAAAA-3′

E1813G
5′-CACCAGCAAGCGACCCAAAAATATCCCCCATGG-3′
5′-CCATGGGGGATATTTTTGGGTCGCTTGCTGGTG-3′

**Western blotting**. We used published *Dicer-KO* MEFs[44]. Cells were incubated in lysis buffer (10 mM Tris-Cl pH 8.0, 150 mM NaCl, 1 mM EDTA, 1% Triton-X, 0.1% SDS, 1% protease inhibitor and 1 mM DTT) for 10 min at 4 °C followed by centrifugation at $18,407 \times g$ for 10 min. The supernatant was collected and total protein concentration was estimated by Bradford assay. Ten micrograms of total protein (per lane) was separated on 4–12% SDS-PAGE (BioRad). Western blotting was performed by standard procedures. The blots were incubated with 1:1000 dilutions of anti-human Dicer (13D6, Abcam) and β-tubulin (E7, DSHB) primary antibodies and 1:10,000 dilution of HRP (horseradish peroxidase)-conjugated anti-mouse IgG secondary antibody (Jackson). The protein bands on the blots were detected using an enhanced chemiluminescent substrate for HRP activity (ECL, Thermo Fisher Scientific).

**Northern blotting**. *Dicer-KO* MEF cells were transfected with published constructs expressing pri-miRNA constructs (*mir-144/451* and either *mir-151* or *mir-199a-1*)[44,45] and either wild-type or mutant hDICER1 construct (WT or R944Q or S1344L or E1813G), as indicated. Transfection was performed in six-well plates using Lipofectamine-LTX reagent (Thermo Fisher Scientific) as per the manufacturer's protocol. Untransfected *Dicer-KO* MEF cells were used as background control. We performed Northern blotting by separating 20 µg of total RNA per lane on 15% polyacrylamide 7M urea gels and transferring onto GeneScreen Plus membrane (Perkin Elmer) using the Trans-Blot SD Semi-Dry Cell (BioRad)[74]. The blots were ultraviolet (UV) crosslinked (Stratagene), baked at 80 °C for 1 h and probed with antisense DNA oligos labeled with γ-[32P]-ATP. The blots were stripped and re-probed to detect multiple miRNAs and loading control (U6). γ-[32P]-ATP-labeled Decade Marker RNA (Thermo Fisher) was used as size standard (10–100 bases). Probe sequences to detect small RNAs are:

miR-144-3p AGTACATCATCTATACTGTA
miR-451-5p AACTCAGTAATGGTAACGGTTT
mir-151-5p TACTAGACTGTGAGCTCCTCGA
mir-151-3p TAACCAATGTGCAGACTACTGTa
miR-199a-1-5p GAACAGGTAGTCTGAACACTGGG
miR-199a-1-3p TAACCAATGTGCAGACTACTGT
U6 snRNA ATTTGCGTGTCATCCTTGCGCAG

**Luciferase sensor assay**. We used published miRNA sensor constructs[45], bearing miRNA target sites downstream of renilla luciferase coding sequence in psiCHECK2 (Promega). Firefly luciferase expressed from psiCHECK2 serves as an internal control. Using Lipofectamine-LTX (Thermo Fisher Scientific) to transfect *Dicer-KO* MEFs in 96-well plates with 160 ng pri-miRNA expression constructs (*mir-151*, *mir-199a-1*) and 40 ng cognate target-containing sensor constructs. Cognate targets were antisense to either 5P or 3P arm of the expressed miRNAs. Fold repression was normalized to parallel assays with non-cognate miRNA (*mir-375*).

**Reporting summary**. Further information on research design is available in the Nature Research Reporting Summary linked to this article.

## Data availability

The somatic mutational data from both the TCGA and MSK-IMPACT cohorts were downloaded from the cBioPortal for Cancer Genomics (http://cbioportal.org/). TCGA datasets were downloaded from the GDC data portal of the National Cancer Institute: TCGA miRNA-Seq data, [https://portal.gdc.cancer.gov/repository?filters=%7B%22op%22%3A%22and%22%2C%22content%22%3A%5B%7B%22op%22%3A%22in%22%2C%22content%22%3A%7B%22field%22%3A%22files.data_format%22%2C%22value%22%3A%5B%22BAM%22%5D%7D%7D%2C%7B%22op%22%3A%22in%22%2C%22content%22%3A%7B%22field%22%3A%22files.experimental_strategy%22%2C%22value%22%3A%5B%22miRNA-Seq%22%5D%7D%7D%5D%7D]; GBM miRNA-microarray data, [http://firebrowse.org/?cohort=GBM]; TCGA endometrial RNA-seq data, [http://gdac.broadinstitute.org/runs/stddata__2016_01_28/data/UCEC/20160128/]. All computational and raw experimental data that support the findings of this study are available within the article, its supplementary information files, and in the Source Data Files. Any other necessary information is available from the corresponding author upon reasonable request. A reporting summary for this article is available as a supplementary information file.

## Code availability

The code for analyses conducted in this study and supplemental results for each of the analyses are available at https://github.com/dicerhotspot/.

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

## Acknowledgements

We thank William Lee for helping with initial mutation calling, Giovanni Ciriello for discussion on gene expression analyses, and Robert Fieldhouse and Debora Marks for contributing to evolutionary couplings analysis. We are grateful to MSK patients and the Marie-Josée and Henry R. Kravis Center for Molecular Oncology for access to MSK-IMPACT profiling data. We thank Kjong Lehmann, Andre Kahles, Gunnar Rätsch, Özgün Babur, Pınar Aksoy, Ed Reznik, Nils Weinhold, Ruomu Jiang, Berkin Elvan for comments. This work was supported by US National Cancer Institute funding of the TCGA Genome Data Analysis Center (U24 CA143840). E.C.L.'s group was supported by the NIH/NIGMS (R01-GM083300), NIH/NHLBI (R01-HL135564) and the Functional Genomics Initiative (FGI), and MSK Core Grant P30-CA008748.

## Author contributions

J.V. conducted most of the computational analyses. W.K.C. analyzed DICER1 mutations. B.A.A. created many conceptual frameworks for the study, and conducted the RNA-seq analysis. S.M. performed experimental tests of DICER1 mutants. A.J.S. helped with initial miRNA-seq analysis and E.D. contributed to pathway analysis. C.S. analyzed evolutionary couplings. N.S., C.S., and E.C.L. supervised the studies and analyzed data. E.C.L. wrote the manuscript with input from co-authors.

## Additional information

**Competing interests:** The authors declare no competing interests.

