## [Peer Review File · Nature Communications]

Reviewers' comments:

Reviewer #1 (Remarks to the Author):

This is a well written report investigating the prevalence and function of RNase IIIa and IIIb domain mutations of DICER1 in cancer. The authors investigate large published data sets as a basis of their discovery of additional mutations, as well as evaluation of functional importance of already reported mutations. They find that their RNase IIIa mutation also leads to miRNA-5p defects, and find that these mutations impact important miRNA families and gene expression.

The main concerns with the paper are the relevance of the findings for cancer. The authors frequently allude to their importance in "progression", and "tumorigenesis," but there is no real evidence to support this. They are extremely rare, and, while some appear to influence miRNA expression and gene expression, there is no enrichment in metastatic disease, nor study of the association with more aggressive endometrial cancer subtypes. For any real insight into their importance in cancer, instead of merely as a rare occurrence, it would be necessary to investigate either their occurrence in premalignant versus malignant tissue, their association with tumor grade, or subtype, or ideally, their association with outcome. All of this should be possible in endometrial cancer at least.

Reviewer #2 (Remarks to the Author):

Aksoy et al investigate recurrent mutations in the DICER1 domains RNase IIIb and RNase IIIa in 30,000 tumor samples, revealing an uncharacterised recurrent mutation in RNase IIIa. A surprising find is that the RNase IIIa S1344L mutation leads to a depletion of 5p-miRNAs similar to RNase IIIb hotspot mutations, although the two domains are processing opposite sides of the pre-miRNA hairpin. This result was validated by functional assays, which looks convincing to me. The authors hypothesise on a structural explanation for the potential coupling of RNase IIIa S1344L to the RNase IIIb catalytic residues. Furthermore, down regulation of target genes of let-7 and miR-17 family 5p-strand miRNAs is shown, although only in 12 cases having all necessary data types and a hotspot mutation.

These results are highly interesting not only for cancer researchers, as they also reveal novel aspects of miRNA biosynthesis. Nonetheless I see several issues with the manuscript that would need to be addressed.

1. The frequency of the hotspot mutations, including the newly identified S1344L, is very low in both cohorts. Therefore, I have a few questions regarding the identification of hotspots and the significance of the observations:

A) How have hotspots been identified, i.e. how are hotspots distinguished from non-hotspot mutations? Is the significance of the recurrence measured and are the p-values corrected?

B) The authors state in the Methods that oncoKB.org and cancerhotspots.org have been used to identify DICER1 mutation hotspots, but I cannot find any S1344L mutation on these websites. I also see no methods/tools to be used on or downloaded from these websites. This needs to be explained in detail. It is completely unclear to me which hotspot detection algorithm has been used.

C) In the Bitbucket link <https://bitbucket.org/armish/5pdep> containing the analysis scripts for the study (link provided in the Code Availability section) I see that cBioPortal was used to download hotspot mutations. This has never been mentioned in the manuscript. Are hotspots identified using cBioPortal and not oncoKB or cancerhotspots.org?

2. Depletion of miRNA-5p species in RNase IIIb hotspot mutant cases is highly significant ($p < 10^{-29}$), as expected. However, no p-value is provided for the cases with RNase IIIa-S1344L mutation. The authors state that (quote) “relatively low 5p abundance” was found in two S1344L cases, but not how significantly this “relatively low” is. Furthermore, why is this effect only seen in 2 cases? If I count correctly there are a total of 9 cases with S1344L. Is this observation still valid when taking all S1344L mutant cases into account? Please provide a p-value, otherwise this cannot be sold as evidence for a similar behavior of S1344L mutants and RNaseIIIb hotspot mutations.

3. Another part I am not fully convinced of is the Identification of evolutionary couplings and the 3D mapping of mutations. But this is likely because the result part is confusing and the methods description is not much more enlightening either. The 3D structure of Fig.5 does not help to explain the proposed interactions and evolutionary couplings. The results of the evolutionary coupling analysis (i.e. results from the EVFold server) are not presented at all, just discussed. The listed result file `EvCouplings_DICER1_RNaseIIIb_with_2eb1.zip` was not part of the provided review material. In summary, the description of both the used method and the obtained results has to be substantially more detailed.

Reviewers' comments:

Reviewer #1 (Remarks to the Author):

This is a well written report investigating the prevalence and function of RNase IIIa and IIIb domain mutations of DICER1 in cancer. The authors investigate large published data sets as a basis of their discovery of additional mutations, as well as evaluation of functional importance of already reported mutations. They find that their RNase IIIa mutation also leads to miRNA-5p defects, and find that these mutations impact important miRNA families and gene expression.

We thank the referee for their positive comments on our study.

Based on the feedback from this and the other referee, and due to personnel changes in the lab during the revision process, we had to undertake a complete overhaul of the computational procedures in order to be able to conduct new analyses. Most of the figures have been changed and many supplementary figures were added, and these had to be built from new analysis pipelines. Major text revisions are highlighted in yellow to draw attention to differences from the original submission, which we recognize was a while ago. The updated manuscript is much more rigorous and robust and we hope that the referees will agree on the contributions we make to cancer genetics and miRNA biogenesis.

The main concerns with the paper are the relevance of the findings for cancer. The authors frequently allude to their importance in "progression", and "tumorigenesis," but there is no real evidence to support this. They are extremely rare, and, while some appear to influence miRNA expression and gene expression, there is no enrichment in metastatic disease, nor study of the association with more aggressive endometrial cancer subtypes. For any real insight into their importance in cancer, instead of merely as a rare occurrence, it would be necessary to investigate either their occurrence in premalignant versus malignant tissue, their association with tumor grade, or subtype, or ideally, their association with outcome. All of this should be possible in endometrial cancer at least.

We appreciate the referee's concerns, since DICER1 hotspot mutations studied in this manuscript are infrequent amongst cancer patients overall. Given the types of analysis that TCGA and MSK-IMPACT data are suitable for, we agree we could not make conclusions regarding their association to "progression" and thus we removed such comments.

On the other hand, we respectfully disagree with the referee regarding functional relevance of DICER hotspot mutations in cancer ("*The main concerns with the paper are the relevance of the findings for cancer...*"). We have shown that known DICER1 hotspot mutations coupled to biallelic inactivating events and are enriched in very specific types of cancer. We have supplemented our paper with statistical analysis to support this, which is reproduced in the independent TCGA and MSK-IMPACT cohorts.

We find it a very biologically meaningful result that out of several dozens of cancer types, diverse types of uterine/endometrial cancers are the only reliably statistically enriched cancer types bearing DICER1 hotspot mutations (sarcoma and sex cord stromal tumor also came out from the IMPACT analysis). No hotspot mutations were seen in numerous common cancer types that have been more deeply sequenced than endometrial/uterine cancers. The depletion of Dicer hotspot mutations in diverse cancers is a reciprocal biological indication that the enrichment in uterine cancers is selectively beneficial to this setting, even though Dicer hotspots are very sporadically observed in diverse cancers. We have updated main Figure 2 with current MSK-IMPACT patient data (now >10,000 more cases), and added several Supplementary Figures (2 and 3) to show the enrichment/depletion analysis of hotspot mutations by cancer type.

Perhaps of greater novelty is the fact that our study has now functionalized an orphan DICER1 mutation with a molecularly defined impact on miRNA biogenesis that is unexpectedly analogous to known DICER1 hotspot mutations. The novel DICER1 RNase IIIa mutation (S1344L) we identify and subject to biochemical and structural analysis is rare, but clearly the target of specific somatic cancer mutations (Q-value = 0.0281). A supplementary figure depicting the four DICER1 mutations now called in the CancerHotspots.org database for statistically significant mutations in cancer has now been added, and includes DICER1-S1344L along with several RNase IIIb hit residues.

Importantly, we note that contrary to the expectations cited by the Referee on clinical correlations for relevant cancer drivers ("*...there is no enrichment in metastatic disease, nor study of the association with more aggressive endometrial cancer subtypes.*"), many established cancer driver genes have comparable mutation frequencies in untreated primary and metastatic tumors (Zehir et al, Nature Medicine, 2017) and do not necessarily show association with tumor grade or patient outcome (Kandoth et al., Nature, 2013). Still, we think it is notable that we have found that

known and novel (S1344L) hotspot+biallelic inactivating events are present in metastatic cases in the MSK-IMPACT patient set, which indicates that these specific mutated Dicer cancer cells were able to seed new tumors. This is significant since the TCGA dataset is composed of primary tumors, and it is implied from at least some experimental tests that Dicer mutant cells are disadvantaged (metastasis requires a distinct set of colonization challenges to be satisfied compared to primary tumorigenesis).

We attempted to conduct further analysis along the suggestion of the referee, using the endometrial cases. The limited number of DICER1 mutant cases restricts the interpretation of clinical associations. Still, a potentially interesting result from the TCGA endometrial cohort is shown below. When comparing the DICER1 hotspot mutants to other bulk DICER1 mutants in non-hotspot residues, we observe a statistically significant shorter overall lifespan (most of the bulk DICER1 cases were still alive in the UCEC clinical data).

We feel these data need to be interpreted cautiously, since the larger cohort of DICER1-wt patients also had shorter overall survival than DICER1-"non-hotspot" mutants. One possible explanation for this is if the aggregate DICER1-wt patients collectively bear many other driver mutations, at least a portion of which also reduce lifespan (there are 10X more patients in this cohort). In this interpretation, the best direct comparison for DICER1-hotspot mutants is other patients with somatic DICER1 mutations.

These tests are what we can do with the available datasets, and we provide them for referee consideration; but we do not feel they are robust enough to include in the paper. We do not wish for this to detract from the major focus of our study, and emphasize that it is focused on leveraging knowledge on miRNA mechanisms to gain new insights in largescale cancer genomics data. In this respect, we believe that we have generated robust genomic and biochemical conclusions.

To further bolster the genomic analysis, we undertook a significant overhaul of the computational pipeline, necessitating a change in authorship. Over the past half year, we have completely redone the miRNA analysis to account for the fact that far greater small RNA datasets are now available in the final PanCan dataset than when the

initial analysis was conducted. In total we updated a comprehensive analysis with nearly 7000 additional samples from the original submission (including more known RNase IIIb hotspot cases and increasing the novel S1344L small RNA datasets from 2 to 5).

We have refined how we conducted this analysis, and provided several new figures, including an informative breakdown of analyses by tumor type. We have also redone all the structural comparisons and the evolutionarily couplings analysis, as motivated by Referee 2. Additional responses are provided to the other Referee and we assume they are available to Referee 1. We hope the Referee appreciates that these collected studies will be of substantial impact for the miRNA biogenesis and cancer communities and will find on balance reason to be supportive of the work.

Reviewer #2 (Remarks to the Author):

Aksoy et al investigate recurrent mutations in the DICER1 domains RNase IIIb and RNase IIIa in 30,000 tumor samples, revealing an uncharacterised recurrent mutation in RNase IIIa. A surprising find is that the RNase IIIa S1344L mutation leads to a depletion of 5p-miRNAs similar to RNase IIIb hotspot mutations, although the two domains are processing opposite sides of the pre-miRNA hairpin. This result was validated by functional assays, which looks convincing to me. The authors hypothesise on a structural explanation for the potential coupling of RNase IIIa S1344L to the RNase IIIb catalytic residues. Furthermore, down regulation of target genes of let-7 and miR-17 family 5p-strand miRNAs is shown, although only in 12 cases having all necessary data types and a hotspot mutation.

These results are highly interesting not only for cancer researchers, as they also reveal novel aspects of miRNA biosynthesis. Nonetheless I see several issues with the manuscript that would need to be addressed.

We thank the referee for their positive comments on our study.

Based on the feedback from this and the other referee, and due to personnel changes in the lab during the revision process, we had to undertake a complete overhaul of the computational procedures in order to be able to conduct new analyses. Most of the figures have been changed and many supplementary figures were added, and these had to be built from new analysis pipelines. Major text revisions are highlighted in yellow to draw attention to differences from the original submission, which we recognize was a while ago. The updated manuscript is much more rigorous and robust and we hope that the referees will agree on the contributions we make to cancer genetics and miRNA biogenesis.

1. The frequency of the hotspot mutations, including the newly identified S1344L, is very low in both cohorts. Therefore, I have a few questions regarding the identification of hotspots and the significance of the observations:

A) How have hotspots been identified, i.e. how are hotspots distinguished from non-hotspot mutations? Is the significance of the recurrence measured and are the p-values corrected?

The overall number of mutations is modest in the overall population of cancer patients, but it is clear from visual inspection that they are recurrent above the background spectrum, and that this has been replicated in the independent TCGA and IMPACT cohorts, which represent distinct types of cancer sequencing (primary tumors vs. treated patients and metastatic cases).

We now note that S1344L has since been added to the CancerHotspots.org database for statistically significant mutations in cancer (Q-value = 0.0281). A supplementary figure depicting the four DICER1 mutations now called in the CancerHotspots.org database is now provided, which recently includes DICER1-S1344 along with several RNase IIIb residues. However, some of the other previously published RNase IIIb catalytic site hotspot mutations are not in the CancerHotspots database, but are clearly recurrent in both TCGA and MSK-IMPACT. We have clearly cited these, and revised the figure to cite the case numbers.

B) The authors state in the Methods that oncokb.org and cancerhotspots.org have been used to identify DICER1 mutation hotspots, but I cannot find any S1344L mutation on these websites. I also see no methods/tools to be used on or downloaded from these websites. This needs to be explained in detail. It is completely unclear to me which hotspot detection algorithm has been used.

We apologize for the confusion, as the project had evolved over time and the text had undergone modifications over several iterations over years. Originally, the novel recurrent mutations were noticed by manual evaluation at an early stage of the TCGA project, and were subsequently subjected to other computational and biochemical tests that convinced us of their functional impact. As more cases accumulated in the TCGA and then the IMPACT datasets, their statistical significance became possible to evaluate. We have made some modification to the text to acknowledge the evolution of recognizing functional significance of S1344L; i.e., that it was not identified in the original Cancer Hotspots analysis (Chang et al, 2016), but is now recorded in a subsequent analysis (Chang et al, 2018). As mentioned, S1344L is on the current cancerhotspots.org database, although it does not contain all of the RNase IIIb hotspot mutations annotated in the literature; these are found in oncokb.org.

We have amended the methods to note which hotspots we have used and their sources. Nevertheless, by presenting the full set of somatic mutations in Figure 1 in both TCGA/IMPACT datasets, it should be clear where the hotspots lie, and we don't have to dwell too much on the history of how we came to recognize this interesting mutation when there were far fewer tumor sequences available. Since the last submission, we have re-evaluated the current MSK-IMPACT data and there more RNase IIIb hotspot mutations than before; as documented in Figure 1 and Supplementary Tables 1 and 2.

As a side note, there are several aspects of the analysis that have been redone over the years. For example, the biallelic alterations of DICER1 were absent in original TCGA calls, such that secondary DICER1 events were identified using a series of in-house comparisons of mutation callers. Eventually these were superseded by the latest iteration of TCGA MC3 MAF 2+ caller pipeline used in the final PanCan analysis. For the final analysis, the mutation calls are aligned with what was used in the PanCan analysis.

C) In the Bitbucket link <https://bitbucket.org/armish/5pdep> containing the analysis scripts

for the study (link provided in the Code Availability section) I see that cBioPortal was used to download hotspot mutations. This has never been mentioned in the manuscript. Are hotspots identified using cBioPortal and not oncokb or cancerhotspots.org?

We appreciate the careful review of the materials that the referee has done. As mentioned, this reflects a methods versioning issue which we have updated in the current revised methods. Hotspot calls refer to alterations in 6 specific amino acids (1 in RNase IIIa, 5 in RNase IIIb) with demonstrated effects on 5p:3p biogenesis. We have overlaid those onto all DICER1 alterations in cBioPortal.

2. Depletion of miRNA-5p species in RNase IIIb hotspot mutant cases is highly significant ($p < 10^{-29}$), as expected. However, no p-value is provided for the cases with RNase IIIa-S1344L mutation. The authors state that (quote) “relatively low 5p abundance” was found in two S1344L cases, but not how significantly this “relatively low” is. Furthermore, why is this effect only seen in 2 cases? If I count correctly there are a total of 9 cases with S1344L. Is this observation still valid when taking all S1344L mutant cases into account? Please provide a p-value, otherwise this cannot be sold as evidence for a similar behavior of S1344L mutants and RNase IIIb hotspot mutations.

To further bolster the genomic analysis, we had to undertake a significant overhaul of the computational pipeline, necessitating a change in authorship. Over the past half year, we have completely redone the miRNA analysis to account for the fact that far greater small RNA datasets are now available in the final PanCan dataset than when the initial analysis was conducted (at that time, there were only 2 S1344L cases in the 3000 small RNA datasets). In total we updated a comprehensive analysis with nearly 7000 additional samples from the original submission (including more known RNase IIIb hotspot cases and increasing the novel S1344L small RNA datasets from 2 to 5, one of which is trans-heterozygous with S1344T). Please note that MSK-IMPACT cohort only has genomic sequence and lacks small RNA data; thus the small RNA data only refers to TCGA cases.

This is germane to the point raised by the referee. First, to clarify, the original statistical test mentioned by the referee refers to an expression-based test for individual miRNA-5p and -3p species. This requires a sufficient number of individual mutant cases of a particular tumor type, so that they express more or less similar miRNA profile, so that changes in miRNA expression are likely to reflect biogenesis rather than tissue type. (The consequence of this is now more evident now that we provided additional analysis by tumor type, see below).

The original analysis produced signal because RNase IIIb hotspot mutants accompanied by biallelic inactivating alterations are dominated by endometrial cancer cases. However we have redone this analysis (scatterplot and bar graph summary of 5p/3p expression shifts) of RNase IIIb by (1) restricting the comparison to endometrial cancer, (2) assigning properly 5p/3p status to all miRNAs, (3) enlarging the set of miRNAs analyzed to include all broadly-conserved loci and a subset of additional well-expressed loci, in total 280 loci. The overall trend is similar to before, but we feel the way the analysis is conducted is more robust.

An expression based comparison will not be reasonable if there are insufficient samples for comparison, and unfortunately, the remainder of the hotspot cases distribute into diverse cancer types but there are only 1-3 cases each per subgroup. For S1344L,

there are 5 patients, but the most is 2 endometrial. Thus we cannot use this strategy to do expression comparisons to generate statistics.

However, as mentioned, we have updated the m'_{53} analysis to include nearly 7000 additional samples, and we have conducted many analyses of this. While a convenient signal-number summary of entire small RNA library, we do expect that it is influenced by the distribution of tissue-specific miRNAs. One thing we did not previously appreciate, but now show by plotting m'_{53} by tumor type, is that several specific tumor types (e.g. most notably ovarian epithelial tumor) show characteristically low overall values, in the presence of wildtype Dicer. However, many RNase IIIb hotspot mutants still fall to the bottom of many individual cancer types, even if they appear "higher" in the aggregate plot with all 10,000 cases, thus demonstrating the effect of DICER1 hotspot alteration. We have added Figures showing these data.

With this highly expanded set of cases, we are able to address the Referee's concern. As a group, non-hotspot DICER1 mutant cases occupy a similar m'_{53} score range as DICER1 wt tumors. On the other hand, RNase IIIb hotspot samples coupled to biallelic inactivating mutations exhibit by far a lower m'_{53} score range. The other RNase IIIb hotspot cases collectively score intermediately, but their range spans the gamut. We interpret that some have inactivating mutations that were not uncovered by genome sequencing, while others have hypomorphic events, and still others might be heterozygous. By contrast, the S1344L cases also have an intermediate m'_{53} metric, but their range is fairly tight. This is consistent with the notion that S1344L is not a direct hit in RNase IIIb catalytic center, but it is one of the most overtly function-altering DICER1 mutants in cancer and all 5/5 TCGA cases are coupled to biallelic inactivating mutations (Figure 1). Moreover, each of the S1344L cases is individually amongst the low-scoring m'_{53} outliers within their individual cancer types (Figure 3). Since these groups are all of sufficient size, we are able to conduct appropriate statistical tests to compare their values and variance amongst the different classes of DICER1 mutants and bulk control tumors. This should address the referee concern.

3. Another part I am not fully convinced of is the Identification of evolutionary couplings and the 3D mapping of mutations. But this is likely because the result part is confusing and the methods description is not much more enlightening either. The 3D structure of Fig.5 does not help to explain the proposed interactions and evolutionary couplings. The results of the evolutionary coupling analysis (i.e. results from the EVFold server) are not presented at all, just discussed. The listed result file `EvCouplings_DICER1_RNaseIIIb_with_2eb1.zip` was not part of the provided review material. In summary, the description of both the used method and the obtained results has to be substantially more detailed.

We recognize the careful reading of the referee and our oversight in not providing the listed result file. Because the analysis had been performed some time ago, and more sequences are available and EVFold package has since been updated, we have redone this analysis. The results are consistent but more compelling than before, to support that S1344 is a highly evolutionarily coupled with F1706, a residue adjacent to the RNase IIIb active site (top 1.5% of all couplings). Moreover, it resides within a patch of highly coupled residues involved in inter-domain interactions between the interface of RNase IIIa and IIIb. We have added new main and Supplementary Figures to illustrate this, provided the underlying EvCouplings output files, and updated methods and text.

Importantly, the cryoEM structure of full-length DICER1 has also since become available since our original submission (Liu Cell 2018), and this is a more relevant structure to inspect for inter-domain contacts rather than modeling based on the old RNase IIIb homodimer (Takeshita JMB 2007). We have redone this analysis and added new figure, which nicely supports that S1344 is physically the closest RNase IIIa residue to the RNase IIIb active site. This remarkable feature can explain how cancer cells (particularly endometrial/uterine cancer cells) exploit a previously unknown aspect of DICER1 catalysis.

Because we have overhauled the scripts for the miRNA analysis, we no longer refer the reader to the bitbucket link. We have substantially updated the methods, provide more of the necessary output files and Supplementary Tables and materials, and now make the new set of scripts available on GitHub.

REVIEWERS' COMMENTS:

Reviewer #1 (Remarks to the Author):

The authors have added significant amounts of data and have addressed previously raised concerns, mainly about over interpreting the findings indicating a functional role of the mutations as leading to a poor prognosis. The manuscript is significantly improved.

Reviewer #2 (Remarks to the Author):

The manuscript by Vedanayagam et al. (formerly Aksoy et al) has been substantially revised. The total number of used samples increased (especially the small RNA data) and several analyses were completely redone. Several of my criticisms have been addressed sufficiently by this re-analysis. In detail:

1. One of my complaints was the lack of detail in method descriptions. This has been addressed and all methods have been described with sufficient detail now.
2. Identification of hotspot mutations and p-values: this has been addressed, mostly because of a larger sample number, which also resulted in a significant p-value (and listing) of S1344L on cancerhotspots.org. Acceptable for me.
3. Analysis of S1344L effects on 5p depletion in only 2 samples: again, the substantially increased sample size helped here. Although a total of 5 samples is still not impressive the 3 new samples support the original claim, as far as I understand Fig. 3F. Definitely an improvement.
4. Evolutionary couplings: has been redone using new cryo-EM structures of full-length DICER1. The new Fig. 5 is a substantial improvement compared to the previous version of the manuscript (specifically 5A). It demonstrates fairly well the close localisation of S1344 and the other RNase IIIb hotspots.

In summary, the paper has substantially improved. The main finding that hotspot mutations in DICER domains RNase IIIb and the newly discovered hotspot in RNase IIIa lead to similar functional consequences, likely explained by close proximity to the catalytic site of RNase IIIb, is much more convincing now.